

# Inter-annual variability of surface ozone at coastal (Dumont d'Urville, 2004-2014) and inland (Concordia, 2007-2014) sites in East Antarctica

5    M. Legrand[1,2], S. Preunkert[1,2], J. Savarino[1,2], M. M. Frey[3], A. Kukui[4], D. Helmig[5], B. Jourdain[1,2], A. Jones[3], R. Weller[6], N. Brough[3], and H. Gallée[1,2]

[1]Université Grenoble Alpes, Laboratoire de Glaciologie et Géophysique de l'Environnement (LGGE), 38000 Grenoble, France

[2]CNRS, Laboratoire de Glaciologie et Géophysique de l'Environnement (LGGE), 38000 Grenoble, France

[3]British Antarctic Survey, Natural Environment Research Council, Cambridge, UK

[4]Laboratoire Atmosphère, Milieux et Observations Spatiales (LATMOS), UMR8190, CNRS-Université de Versailles Saint Quentin, Université Pierre et Marie Curie, Paris, France

[5]Institute of Arctic and Alpine Research (INSTAAR), University of Colorado, Boulder, USA

[6]Alfred Wegener Institut für Polar und Meeresforschung, Bremerhaven, Germany

*Correspondence to*: M. Legrand (legrand@lgge.obs;ujf-grenoble.fr)

**Abstract.**

Surface ozone has been measured since 2004 at the coastal East Antarctic site of Dumont d'Urville (DDU) and since 2007 at the Concordia station located on the high East Antarctic plateau. This paper discusses long-term trends, seasonal and diurnal
cycles, as well as inter-annual summer variability observed at these two East Antarctic sites. At Concordia, near surface ozone data were complemented by balloon soundings and compared to similar measurements done at the South Pole. The DDU record is compared to those obtained at the coastal site of Syowa also located in East Antarctica, as well as the coastal sites of Neumayer and Halley, both located at the coast of the Weddell Sea in West Antarctica. Surface ozone mixing ratios exhibit very similar seasonal cycle at Concordia and the South Pole. However, in summer the diurnal cycle and the vertical
distribution of ozone above the snow surface are different at the two sites with a drop of ozone in the afternoon at Concordia and not at the South Pole, and a far well-mixed rich ozone layer within the lower 250 m at Concordia than at the South Pole during sunlight hours. These differences are related to different solar radiation and wind regimes encountered at these two inland sites. DDU appears to be the coastal site where the impact of the late winter/spring bromine chemistry is the weakest, but where the impact of $NO_x$ snow emissions from the high Antarctic plateau is the highest. The highest impact of the
bromine chemistry is seen at Halley and Neumayer, and to a lesser extent at Syowa. These three sites are only weakly



impacted by the NO$_x$ chemistry and the net ozone production occurring on the high Antarctic plateau. The differences in late winter/spring are attributed to the abundance of sea-ice offshore the sites whereas those in summer are related to the topography of East Antarctica that promotes the katabatic flow bringing oxidant-rich inland air masses to the site. There appears to be a decreasing trend in summer at the two East Antarctic sites of Concordia and DDU over the most recent period (2004/07-2014). Further researches including continuing monitoring are needed at these two sites to better separate effect of synoptic transport from possible change of NO$_x$ snow emissions in response to change of the stratospheric ozone layer.

## 1. Introduction

In remote environments such as the Polar Regions, natural processes, synoptic transport and/or downward transport from the stratosphere, are the primary processes influencing the levels of tropospheric ozone. Consequently, ozone data from these high latitudes were first thought to be of interest in assessing ozone background levels and trends at hemispheric scales. In the Northern hemisphere, the longest polar ozone record is from Barrow (71°19'N, 156°36'W, 8 m above sea level (asl)), showing a mean annual increasing trend of $0.05 \pm 0.08$ ppbv yr$^{-1}$ (Helmig et al., 2007a) between 1975 and 2005. An even lower increasing trend of $0.02 \pm 0.09$ ppbv yr$^{-1}$ (Helmig et al., 2007a) is reported for the longest polar record of the Southern hemisphere available from the South Pole (1975-2005). Whereas these annual trends that remain far lower than those reported at remote sites located at mid latitudes, and are not significantly different from zero at the $P > 95\%$ confidence level, the examination of seasonal trends sometimes revealed more significant changes in both polar regions (see the review from Helmig et al., 2007a). Interpretation of these seasonal trends are however not simple since they are controlled by several recently discovered local processes involving snow or sea-ice coverage. For instance, an intriguing observation of sudden decreases of ozone in the Arctic in spring was reported in the pioneering studies from Oltmans (1981) and Bottenheim et al. (1986). The most important identified mechanism for this phenomenon (also detected at the coastal Antarctic site of Neumayer, Crawford et al. (2001), Frieß et al. (2004)) is the reaction of ozone with halogens originating from oceanic regions covered by sea-ice (Oltmans et al., 1989; Barrie and Platt, 1997). Future changing sea-ice conditions are thus expected to influence ozone trends in spring in polar regions.

Another process that influences ozone in the polar atmospheric boundary layer is related to the photo-denitrification of the snowpack. This denitrification resulting in NO$_x$ emissions in summer was identified over Greenland (Honrath et al., 1999) and Antarctica (Jones et al., 1999, 2000, 2001; Davis et al., 2001). In the case of Antarctica, the evidence for the subsequent production of ozone was first discussed for the South Pole (Crawford et al., 2001). Since the ozone column density controls the UV irradiation at the ground level, Jones and Wolff (2003) suggested that the development of the stratospheric ozone hole has enhanced NO$_x$ snow emissions and ozone photochemical production in the South Pole boundary layer in the early 80's. According to this hypothesis, the expected future slow recovery of the stratospheric Antarctic ozone layer would lead to a decreasing rate of ozone production in the lower troposphere over the Antarctic plateau, at least in spring.



In parallel to these year-round surface ozone studies, several research campaigns dedicated to the study of atmospheric oxidants were conducted, first in January-February 1994 at the coastal site of Palmer (64°46'S, 64°03'W) in the West Antarctic peninsula within the framework of the SCATE project (Sulfur Chemistry in the Antarctic Troposphere Experiment, Berresheim and Eisele (1998)). Later on, the CHABLIS (Chemistry of the Antarctic Boundary Layer and the Interface with Snow, Jones et al. (2008)) project was conducted at Halley from January 2004 to February 2005. In addition to the $HO_x$ chemistry, CHABLIS also targeted the halogen chemistry. The oxidative properties of the atmosphere inland Antarctica were mainly investigated at the South Pole through the ISCAT (Investigation of Sulfur Chemistry in Antarctica) projects in 1998 (Davis et al., 2001) and 2000 (Davis et al., 2004), followed by the Antarctic Tropospheric Chemistry Investigation (ANTCI, Eisele et al. (1998)) project in 2003. Studies of oxidants in the atmospheric boundary layer including ozone were also carried out in summer above the West Antarctic Ice Sheet (WAIS) as part of scientific overland traverses (Frey et al., 2005) and deep ice core drilling at WAIS Divide (Masclin et al., 2013).

With the exception of the Syowa site, for which the ozone year-round record exists since 1997 (Helmig et al., 2007a), the ozone documentation in the East part of Antarctica only started in the last decade with measurements initiated in 2004 at the coastal East Antarctic site of Dumont d'Urville (DDU) and in 2007 at the Concordia station located on the high East Antarctic plateau (Legrand et al., 2009). This latter study revealed some particularities of East Antarctica that influence surface ozone levels such as frequent occurrence of katabatic flow and less sea-ice coverage offshore the coastline. That represented one motivation of the OPALE (Oxidant Production over Antarctic Land and its Export) initiative. The first OPALE campaign took place at DDU in 2010/2011 and results were published in 4 companion papers at the Journal of Geophysical Research (see Preunkert et al., 2012). Data obtained during the second campaign conducted at Concordia in 2011/2012 are discussed in a set of papers in a special issue for Atmospheric Chemistry and Physics of which this is one.

With the aim to better understand interactions between transport (vertical stability of the atmospheric boundary layer, synoptic transport) and chemical (XO and $NO_x$ chemistry) processes that control the surface ozone Antarctic budget (and its future response to global climate change), this paper examines the most recent ozone records available from Antarctica. Examination focuses on data recently gained at the two East Antarctic sites of DDU (2004-2014) and Concordia (2007-2014) but also includes a comparison with those from the South Pole and other coastal sites located both in East and West Antarctica. Finally, this paper briefly shows synoptic conditions with respect to ozone levels that were encountered during the two OPALE campaigns conducted at DDU in summer 2010/2011 and Concordia in summer 2011/2012.

## 2. Sites and methods

At the coastal site of DDU (66°40'S, 140°01'E, 40 m asl, Fig. 1), ozone measurements started in 2004. Ozone monitoring was also initiated in 2007 at the inland site of Concordia (75°06'S, 123°20'E, 3233 m asl) located near Dome C (DC, 1100 km away from the nearest coast of East Antarctica, Fig. 1). At both sites, measurements have been made continuously with a UV absorption monitor (Thermo Electron Corporation, Franklin, MA), model 49 I at Concordia and 49 C and DDU. Zero



determinations and gas standard calibrations were made one time per month. The data collected at 15-s intervals are here used as hourly, daily and monthly averages. At Concordia, where the instrument and the inlet of the sampling line are located at 17 m above the ground level, measurements were occasionally disturbed for a few hours when the wind was blowing from the power supply building of the station (sector 20-50°W). In addition, under very low wind speed conditions

($< 2$ m s$^{-1}$) measurements were disturbed for a few minutes when vehicles were leaving the station around 8:00 and 14:00 or back from work at 12:00 and 19:00. Such events can be easily identified by fast decreases of the ozone level in the 15-min record and were discarded when calculating hourly averaged values. The effect of such sporadic contamination had however a limited effect on daily values. Among a total of 2687 daily values, sporadic contamination took place over 426 days. The removal of the corresponding 15-min values led to correction of daily values by more than 2 ppbv for 49 days and more than

1 ppbv for 106 days. At DDU, in some occasions the wind blew from the main combustion points of the station located between 0 and 90°E and between 310° and 340°E causing the plume of the station power supply to impact the ozone sampling. These data were removed from the record. Among 3870 daily values, corrections were done for 1510 daily values (135 values with a correction exceeding 1 ppbv, 28 values with a correction ranging from 2 to 7 ppbv).

DC and DDU surface ozone data presented in this paper will be discussed and compared to those available at several other

Antarctic sites. In particular, the DC record will be compared to data from the South Pole (89°59'S, 2850 m asl). The DDU record will be compared to those obtained at the coastal site of Syowa (SY), also situated in East Antarctica (69°00'S, 39°35'E, 29 m asl), as well as the coastal sites of Neumayer (NM, 70°39'S, 8°15'W, 42 m asl) and Halley (HA, 75°33'S, 26°32'W, 30 m asl), both located at sea level in the Weddell Sea sector (West Antarctica, Fig. 1). Data from SP, SY, NM, and HA were retrieved from the WMO World Data Centre for Greenhouse Gases (http://ds.data.jma.go.jp/gmd/wdcgg/cgi-

bin/wdcgg) and updated with data from individual investigators. More details on data from the South Pole can be found in McClure-Begley, A., Petropavlovskikh, I., Oltmans, S., (2014) NOAA Global Monitoring Surface Ozone Network.1973-2014. National Oceanic and Atmospheric Administration, Earth Systems Research Laboratory Global Monitoring Division. Boulder, CO. (http://dx.doi.org/10.7289/V57P8WBF). Details on ozone data from Syowa can also be found at http://ds.data.jma.go.jp/gmd/wdcgg/wdcgg.html.

At Concordia, surface ozone data were complemented by investigation of the vertical ozone distribution in summer by using tethered balloons. Electrochemical concentration cell (ECC) sondes (ZECC, EN-SCI Corporation) and PTU sondes (RS92-SGPG Vaisala) interfaced to RSA921-OIF92 (Vaisala) for remote data transfer were used to document the vertical ozone distribution and meteorological conditions. Prior each flight, ECC sondes were calibrated with the Ozoniser/Test Unit TSC-1 provided by Vaisala. Helium-filled (2 m$^3$) tethered balloon (latex Totex, 1000 g) were deployed. Ascent rates typically

ranged from 0.2 to 0.8 m s$^{-1}$, depending on the operator. Given the 25 s response time of the ECC sonde, delivered ozone data are delayed causing a upward shift of the ozone profile by up to 20 m. Vertical profiles were obtained from the end of December 2009 to 15 January 2010 and from 11 December 2011 to 9 January 2012. Under low wind conditions, balloons were able to reach heights up to 1010 m above ground level (agl) (i.e. 4243 m asl). That permitted to document the vertical ozone change from the ground to above the planetary boundary layer (PBL), which grows to several hundreds of meters in





the afternoon. High wind speeds (> 5 m s$^{-1}$) often prevented the balloon to reach more than 200 m above the ground; collected data however still proofed useful to document the change of ozone from the ground to above the thin PBL encountered in the morning and at night. In this way, the vertical distribution of ozone was documented during the afternoon for 10 days and during night/early morning for 7 days.

## 3. Results and discussion of East Antarctic ozone records

Daily and monthly averaged mixing ratios of ozone at Concordia and DDU are shown in Fig. 2 and Fig. 3, respectively. At Concordia, monthly averaged mixing ratios were calculated from February 2007 to September 2014 except for December 2008 and January 2009 over which very limited data were available due to a long breakdown of the analyser (Fig. 2). Furthermore, we do not calculate the monthly mean when a significant fraction of data was missed (30% in December 2007, 36% in January 2011, and 32% in December 2013). At DDU, all monthly averaged mixing ratios are available from March 2004 to October 2014 except in December 2011 when 32% of data are not available (Fig. 3). At both sites monthly means are the highest in mid-winter, in July (34.3 ± 0.7 ppbv at Concordia, 34.2 ± 1.0 ppbv at DDU) and August (33.5 ± 1.6 ppbv at Concordia, 33.5 ± 1.8 ppbv at DDU). The summer minimum is observed in February at Concordia (20.7 ± 2.3 ppbv) and in January (16.5 ± 3.7 ppbv) at DDU. The overall seasonal cycle at Concordia and DDU, characterized by a winter maximum, is thus consistent with what is expected in such remote regions with a winter accumulation of ozone transported from other regions in the darkness, followed by photochemical destruction in spring and summer. At Concordia, however, the decreasing trend from July to October is interrupted by an increase in ozone levels in November, leading to the presence of a secondary maximum (Fig. 2). At DDU, the November secondary maximum is not detected, but at that time, daily mean ozone values often exceed those observed in October (Fig. 3). In the following we focus discussions on this summer time period at Concordia (Sect. 3.1) and DDU (Sect. 3.2). The patterns of the ozone seasonal cycle at Concordia and DDU are then compared and discussed with those observed at other Antarctic sites in Sect. 4 and Sect. 5, respectively.

### 3.1. Concordia

### 3.1.1 Interannual summer variability at Concordia

The observation of a secondary maximum of ozone in November-December at inland Antarctic sites was first reported for the South Pole by Crawford et al. (2001) and attributed to a photochemical production induced by the high NO$_x$ levels in the atmospheric surface layer that are generated by the photo-denitrification of the Antarctic snow-pack (Davis et al., 2001). At DC, a secondary maximum was already reported by Legrand et al. (2009) in November-December 2007, suggesting that photochemical production of ozone in summer concerns a large part of the Antarctic plateau. As seen in Fig. 2, the November-December secondary maximum is clearly detectable in each of the 8 years of the record at Concordia. Also, the



scatter of daily mean values at DC becomes far larger in summer (3.5 ppbv in November, 5.2 ppbv in December, and 4.3 ppbv in January) than over the rest of the year (from 2.3 ppbv in February and 2.0 ppbv in October down to 0.7 ppbv in July). In general, as seen in Fig. 4, a large daily variability is observed from mid-November to mid-January. At that time, ozone levels are enhanced in air masses having travelled over short distances during the last 5 days prior their arrivals at

Concordia. In contrast, prior and after this summer period, ozone values remain unchanged in response to change of the travel distance of the air mass arriving at the site (Fig. 4).

Even when air masses arriving at Concordia had travelled far more than 1000 km (i.e. the distance from the coast), they have rarely passed over the ocean during the 5 days before their arrival at the site. For instance during summer 2012/2013, only in two occasions (14 November 2012 and 5 March 2013) the air mass had passed over the ocean 2 to 3 days prior to its arrival

at Concordia as recognized in Fig. 4 by the long backward trajectories. The same is true for summers 2009/2010 and 2011-2012 reported in Fig. 5. Only 2 marine transport events occurred during summer 2009-2010 (15 and 19 November, Fig. 5a) and 4 during summer 2011-2012 (15, 21, and 24 December, 12-14 January, Fig. 5c). Note that this later event that is characterized by the longest trajectories (6500 km) encountered over the 8 years of the record coincides with a drop of ozone well below 20 ppbv.

Short trajectories generally correspond to air masses having mainly travelled above the high Antarctic plateau. As seen in Fig. 5, ozone levels are particularly high when the air masses remained above 3200 m elevation. As an example, the fast increase of ozone seen in 2009 from December 21 (around 27 ppbv) to December 23 (around 34 ppbv, see the grey arrow on Fig. 5a) corresponds to an enhancement of the fraction of time spent by the air mass above 3200 m elevation (24% of time from 20 December 20:00 to 22 December 8:00 to 98% of time from 22 December 8:00 to 24 December 8:00), whereas the

mean length of the 5-day backward trajectories remaining similar over the two periods (2055 km and 1960 km, respectively). Note also that, as seen in Fig. 5c, the main OPALE campaign from 19 December 2011 to 9 January 2012 missed the period of high photochemical activity in the beginning December (with hourly ozone values as high as 45 ppbv, i.e. the highest since 2007, not shown). Fig. 5 clearly shows that, in contrast to 2009/2010, the second part of December 2011/2012 was characterized by air masses having travelled over rather long distances during the last 5 days prior their arrivals at

Concordia.

As seen in Table 1, even at the monthly scale, the importance of air mass travel distance on summer ozone levels is still detectable. For instance, for the second half of November, the lowest value in 2013 (28.6 ppbv) corresponds to the longest mean trajectory (3180 km) contrasting with values close to 33.8 ppbv in 2007 when the trajectories were among the shortest (2572 km on average). The same tendency is observed for December with the lowest mean value in 2010 (26.6 ppbv for

mean trajectory of 2852 km) and highest mean value in 2012 (30.8 ppbv for 2430 km). For the first half of January, the lowest value in 2012 (22 ppbv) corresponds to the longest mean trajectory (3196 km) contrasting with values close to 27.5 ppbv in 2010 when the mean length of trajectories was the shortest (2286 km).

The preceding discussions suggests that the surface ozone mixing ratio at Concordia in summer is strongly influenced by the synoptic origin of the air masses, highest ozone values being observed when transport during the 5 days prior to its arrival



was over the highest part of the Antarctic plateau. These transport conditions have favoured the photochemical production and/or accumulation of ozone related to snowpack emissions of $NO_x$ in the air mass arriving at the site.

### 3.1.2. Diurnal changes at the surface and in the lower troposphere in summer at Concordia

Based on observed mixing ratios of NO and $RO_2$ at Concordia (Table 2), an integrated local ozone production from the
reaction of NO with $RO_2$ of 4.7 ppbv per day was calculated by Kukui et al. (2014). As seen in Fig. 6 (c and d), even considering the loss reactions of ozone with OH, $HO_2$ and NO (neglected by Kukui et al. (2014), the ozone net production still reaches 4 ppbv per day. In the following we examine whether such a calculated ozone production rate of a few ppbv per day is consistent with observations. Referring to the influence of synoptic transport conditions on the day-to-day ozone variability discussed in Sect. 3.1.1, examination of trend over several days to estimate a mean local ozone production rate
requires the search for a period of several days over which the transport pattern was not modified. As seen in Fig. 5a, from 27 to 30 November 2009 backward trajectories indicate transport of air masses consistently from the high Antarctic plateau (100% of time above 3200 m asl), and over these 4 days surface wind direction remained unchanged (Fig. 7). An overall increase of around 4 ppbv in surface ozone can be observed (i.e. a mean increase of 1 ppbv per day). Even estimated over a period with a slow synoptic transport, this ozone increase of 1 ppbv per day remains well below the expected local
photochemical production. In fact, as seen in Fig. 7, the overall 4-day trend is interrupted each afternoon by a decrease of ozone mixing ratios by a few ppbv. As previously discussed by Legrand et al. (2009), this ozone decrease in the afternoon was attributed to an increase of the height of the PBL in response to the enhancement of the sensible heat flux in the afternoon at that site. Therefore, the local ozone production can only be evaluated from observations during the few hours before 11:00 when the PBL is still thin (less than 100 m).
To better decipher the effect of ozone photochemical production and change of the vertical mixing over the course of the day, in the following we examine the vertical distribution of ozone obtained from some tethered balloons soundings in summer 2009/2010 and 2011/2012. In the afternoon, most of the time the vertical ozone distribution at Concordia showed strong variations in the lowest 1000 m of the atmosphere with highest mixing ratios within the lowest hundreds m above the snow surface. The case of 4 January 2010 is reported in Fig. 8a when the ozone mixing ratio approached 34 ppbv in the
lower 300 m and decreased steeply by more than 12 ppbv within the next upper 100 m. As shown by the vertical profile of potential temperature (Fig. 8b), this steep vertical change of ozone is clearly related to the presence of a well-mixed (neutral) layer between the snow surface and 300 m above the ground (agl) and a very stable and stratified layer above. Over the 10 days over which flights reach high enough elevations, the thickness of the ozone rich PBL reached in the afternoon $270 \pm 85$ m.
At night, the vertical profiles of the potential temperature showed a very thin (less than 70 m) and stable PBL (see Fig. 8d) contrasting with the deeper and well-mixed daytime PBL. The change generally takes place after 17:00 when surface cooling generates negative buoyancy near the surface and a new boundary layer develops. This nightime boundary layer is more





stratified than the previous daytime boundary layer, which at that time is no more connected with the surface and is now called the residual boundary layer. The residual boundary layer is generally of several hundreds of m depth as seen in Fig. 8d. Its thickness and ozone content sometimes decreased over the course of the night. These changes of the residual boundary layer are somewhat variable likely due to change of wind conditions in these layers.

The onset of solar radiations in the early morning promotes the development of a well-mixed boundary layer. Its thickness is initially small (less than 50 m at 8:00, see Fig. 8f) and reaches a maximum of several hundreds of m around 16:00 in the afternoon as a consequence of the enhancement of radiations and the subsequent increase of the sensible heat flux. As seen in Fig. 9, the growth of the PBL over the course of the day is associated with a change in the ozone vertical distribution. Ozone distribution shows a homogeneous mixing ratio within the entire boundary layer. For the examples reported in Fig. 9,
we will examine if the ozone content of the boundary layer and its change from early morning to mid-afternoon are consistent with a photochemical ozone production in the range of a few tenths of ppbv per hour as calculated for Concordia (see Fig. 6c). As seen in Fig. 9 (a and b), on 31 December 2009 the height of the boundary layer close to 278 m agl for the balloon flight of 14:53 was enhanced to 400 m agl at the next flight (15:31). The ozone mixing ratio that exhibits a homogeneous value within the entire boundary layer during the two soundings, was decreased from 20.0 ppbv (at 14:53) to
19.4 ppbv (at 15:31). Since the decrease of the ozone mixing ratio between the two flights is simultaneous to an increase of the thickness of the well mixed boundary layer, we have examined the ozone inventory in the lower 467 m agl (i.e. the elevation above which ozone mixing ratios remained unchanged between the two flights, Fig. 9a). It is found that the ozone inventory was higher during the second flight than the first flight, the difference corresponding to a net production of 0.35 ppbv over 38 min. The corresponding hourly increase of 0.55 ppbv is consistent with the calculated photochemical
production (on average 0.3 ppbv hr$^{-1}$, Fig. 6c) during sunlight hours of the day at Concordia. Note that comparison of the change over time of the inventory within the entire boundary layer with photochemical ozone production rates calculated using NO mixing ratio observed near the surface (as reported in Table 2) is justified since, at Concordia, vertical NO profiles indicate also homogeneous mixing ratios with gradients remained lower than 10 pptv throughout the afternoon (12:00-18:00 local time), at least within the lower 100 m of the atmosphere as discussed by Frey et al. (2015).

Also interesting is the example reported in Fig. 9 (c and d), showing a slight increase of ozone during mid-morning followed by a strong decrease in the afternoon. The average ozone increase by 0.12 ppbv from 0 to 130 m agl between the flight launched at 10:33 and the one at 11:24 can be attributed to a photochemical production close to 0.14 ppbv per hour having acted within a boundary layer whose the thickness remained closed to 130 m. In the following we examine if the observed drop of ozone from 33 ppbv in the morning to around 27 ppbv at 13:38 is due to the growth of the PBL. The flight at 13:38
only documented the lowest 240 m agl and the potential temperature profile (Fig. 9d) suggests that, at that time the top of the PBL was already higher than 240 m agl. The height of the PBL simulated by the MAR model suggests a PBL height of 140 m and 220 m for the two first flights and of 490 m for the flight launched at 13:38. Detailed on the MAR model are given in this issue (Gallée et al., 2015a). In the following calculations, we have assumed a thickness of the PBL of 400 m at 13:38 (instead of 490 m simulated by MAR) to account for an overall tendency of the MAR model to overestimate the height of the




PBL by a few tens of m at that time of the day (Gallée et al. 2015a). The ozone inventory at 10:33 within the lower 400 m was estimated by extrapolating the vertical gradient of ozone observed above 200 m agl during flight at 10:33 (red curve in Fig. 9c). Assuming a well-mixed ozone level within the PBL at 13:38, an unchanged ozone inventory within the lower 400 m between 10:38 and 13:38 would imply an ozone mixing ratio of 26.3 ppbv at 13:38. If accounting for a mean hourly ozone production of 0.3 ppbv, the 3 hour lag between the first and third sounding leads to a mixing ratio of 27.2 ppbv within the PBL at 13:38. This value is very consistent with the mean mixing of 27.4 ppbv observed within the lower 240 m during sounding at 13:38. Though the thickness of PBL during flight launched at 13:38 is not accurately known, it can be concluded that the large ozone decrease seen between 10:33 and 13:38 (from 33 to 27.4 ppbv) is mainly related to the growth of the PBL. This example thus confirms that the decrease of ozone mixing ratio often detected near the surface in the afternoon (Fig. 7) is related to the increase in the thickness of the boundary layer leading to dilution that counteracts the local ozone photochemical production. It also explains why at Concordia the net accumulation of ozone photo-chemically produced over the course of the day that may reach 4 ppbv over 24 hours is never reached.

## 3.2. DDU

The presence of a rather high ozone mixing ratios in November-December at DDU was already observed by Legrand et al. (2009) and attributed to frequent occurrence of downslope transport to the site of oxidant-rich air masses originating from the high plateau.

As shown in Fig. 10, the ozone levels at DDU in summer are strongly dependant on the origin of air masses reaching the site. Ozone mixing ratios drop when the air masses had travelled most of time over the ocean during the last 5 days prior their arrival at the site. In contrast, highest ozone levels are observed in air masses having never travelled over the ocean and that remained 2 to 3 days above 3000 m asl over inland Antarctica. A particularly good example of the effect of the air mass origin on ozone levels is shown in Fig 10 (a and b). This 2005/2006 summer started with a November month characterized by an absence of marine influence until in the morning of 27 November when a marine event started. This marine event lasted in the afternoon of 29 November and was followed by a time period with air masses coming from the high East Antarctic plateau (from 29 November to 7 December in the morning). This well-marked change of air mass origin was accompanied by a large increase of the ozone mixing ratio by 10 ppbv (from 18.7 ppbv over 27-29 November to 28.5 ppbv during first week of December). As seen in Fig. 10 a and b, during the second week of December air masses came neither from the ocean nor from the inland plateau. These air masses that had travelled over the coastal region of the Antarctic continent during that time contained similar levels of ozone than the marine transport (19.4 ppbv instead of 18.7 ppbv). This similar level of ozone in air masses having a pure marine origin or having travelled over margin region of the Antarctic continent over is consistent with the pioneering study from Frey et al. (2005) that showed that photochemical ozone production is not large over inland Antarctic regions located below around 2500 m. A more recent study conducted in the interior of West Antarctica at WAIS Divide (1766 m asl) also showed that the local ozone production is too small to explain



the observed ozone variability and that surface ozone in summer are systematically linked to rapid (< 3days) transport off the East Antarctic Plateau via katabatic outflow (Masclin et al., 2013). The preceding sequence of changing air mass origin (ocean, Antarctic plateau above 3000 m asl, and coastal regions of the Antarctic continent) was encountered again from 14 to 27 December, with ozone levels of 16.7 ppbv between 14 December in the evening and 17 December in the morning (marine

origin), 23.6 ppbv between 17 December and 21 December (continental origin), and 17.6 ppbv from 21 to 27 December (coastal origin). Finally December ended with another marine event during which ozone levels dropped to 12.7 ppbv.

In contrast to conditions encountered in summer 2005/2006, during the OPALE campaign (26 December 2010 to 14 January 2011), less well-defined changes of transport conditions were observed. For instance, a pure marine regime only occurred for a few hours on the evening of 12 January followed by the distinct continental conditions for 2 days (12 to 14 January). In

fact, as shown by Preunkert et al. (2012) the most dominant regime during the campaign was air masses transport from the coastal regions of the Antarctic continent. In spite of these mixed conditions, in their study of OH radicals Kukui et al. (2012) were able to identify the importance of the production of OH from $RO_2$ recycling in air masses arriving preferentially from inland Antarctica compared to those having been partly in contact with the ocean.

In contrast to the case of Concordia discussed in the preceding Sect. 3.1.2, the examination of the hourly record at DDU does

not reveal a clear diurnal cycle of ozone. As can be seen in Fig. 6a, a mean local photochemical production close to 0.3 ppbv $hr^{-1}$ is calculated during sunlight hours of the day at DDU. As seen in Table 2 the quite similar ozone photochemical production calculated at Concordia and DDU results from a 4 times lower mixing ratio of NO at DDU compared to DC that is compensated by larger $RO_2$ concentrations ($3.4 \times 10^8$ molecules $cm^{-3}$ at DDU instead of $1 \times 10^8$ molecule $cm^{-3}$ at Concordia). The absence of a detectable increase of ozone at DDU over the course of the days may be related to either an

overestimation of the production or an underestimation of sinks. An overestimation of the production cannot be ruled out since NO were estimated from $NO_2$ measurements done at DDU by applying the "extended Leighton" relationship (Grilli et al., 2013). As discussed by Frey et al. (2015), at DC the use of this relationship indicates that observed $NO_2$ mixing ratios are higher than expected with respect to observed NO ones. If it is still the case in continental air masses reaching DDU, the ozone production from NO calculated on the basis of $NO_2$ observations would be overestimated. The estimated net ozone

reported in Fig. 6 neglected dry deposition at the two sites. Whereas this assumption is likely correct for the cold snowpack present at DC, it would not necessarily be correct at DDU given the nature of surrounding surface at the site in summer (open ocean, rocks, soils and wet snow) on which larger ozone dry deposition is expected (see Helmig et al. (2007 b) for a review of ozone dry deposition).

## 4. Comparison between the South Pole and Concordia

As already mentioned in Sect. 3.1, the seasonal cycle of ozone at Concordia characterized by a maximum in July followed a decrease until October and the occurrence of a secondary maximum in November-December, is similar to what is observed at the South Pole. As seen in Fig. 11, monthly mean ozone mixing ratios observed near the surface in November and





December at the South Pole slightly exceed those observed at DC. Over the 2007-2014 period, the South Pole experienced in November and December 22 days with daily mean mixing ratios higher than 40 ppbv, compared to 1 day at Concordia, and 100 days with daily mixing ratios higher than 35 ppbv compared to 35 days at Concordia. The cause of higher ozone values observed near the ground at the South Pole compared to those at Concordia is not simply related to the fact that solar

radiations needed to photo-chemically produce ozone act during 24 hours at the South Pole and not at Concordia. Indeed, the two inland sites are expected to experience rather similar local photochemical ozone production, with 0.13-0.20 ppbv hr$^{-1}$ over the entire course of the day at the South Pole (Chen et al., 2004) and from 0.05 ppbv hr$^{-1}$ at midnight to 0.27 ppbv hr$^{-1}$ during sunlight hours at Concordia (Kukui et al., 2014; see also Fig. 6c). As discussed below, the higher resulting near surface ozone values at the South Pole are primarily related to the difference in the dynamic of the lower atmosphere rather

than to the efficiency of the photochemistry. Notably, the South Pole also experiences more days with low values than Concordia, i.e. 34 days with daily mixing ratios lower than 22 ppbv compared to 13 days at Concordia. The larger number of days with low ozone levels at the South Pole than at Concordia is consistent with more frequent synoptic conditions bringing air mass masses having travelled mainly below 2500 m elevation to the site at the South Pole (Neff et al., 2008) than at Concordia.

The examination of the vertical distribution of ozone in summer at Concordia during sunlight hours and at the South Pole reveals several interesting differences. Concordia and the South Pole experience enrichment of ozone within the lowest few hundred meters above the ground with respect to levels above that are more typical for background conditions (around 20 ppbv, Helmig et al. (2008)). As reported by Oltmans et al. (2008), on a total of 215 summer balloon profiles at the South Pole between 1991 and 2005, a quarter of profiles had no significant vertical gradients and a rather low ozone value near the

surface (21 ppbv). Though fewer vertical profiles are available at Concordia, we never observed such an absence of a ozone vertical gradient. This absence of days without ozone vertical gradients at Concordia is consistent with our previous observation of fewer days with near surface mixing ratios lower than 21 ppbv at that site compared to the South Pole.

When present, the vertical gradients are quite different between Concordia and the South Pole. As seen in Table 3, the enhanced ozone layer sometimes appears deeper at Concordia in the afternoon (up to 400 m) than at the South Pole (from 60

to 200-250 m, Helmig et al. (2008); Oltmans et al. (2008)). Furthermore, within the enhanced layer ozone, the ozone mixing ratios are more homogeneous at Concordia than at the South Pole. In fact, at the South Pole ozone is only homogeneously distributed within the lowest tens of meters above the ground, the layers above always showing a steady ozone decrease down to the values observed within free troposphere layers (Table 3). At Concordia the transition between high ozone values near the surface and background free tropospheric levels is often less than 100 m. It therefore appears that the vertical

stability of the enriched ozone layer is very different at the two sites. In contrast to conditions experienced at the South Pole where there is no diurnal cycle, at Concordia during sunlight hours the temperatures are warmer just above the snow surface than above. This leads to a vanishing of the stratification of the lower atmospheric layers at Concordia during sunlight hours. Furthermore, the role of large-scale dynamical processes, for example the subsidence in the polar cell and the subsequent adiabatic warming of the air, increasing with altitude, is relatively more important in the absence of diurnal cycle and this



favours the stratification of the atmosphere. Finally, the fact that the South Pole is situated on a slope favours inversion winds and mechanical turbulence and this somewhat limits the building of a strong stratification. More details on these different conditions can be found in Neff et al. (2008) and Gallée et al. (2015b). Finally note that at Concordia the stratification of the lower layers occurred only at night but the photochemistry is not efficient at that time of the day.

**5. Comparison between the coastal sites Neumayer, Halley, Syowa, and Dumont d'Urville**

While the seasonal cycles at the 4 coastal sites are similar, and all display seasonal maxima close to 35 ppbv in July (Fig. 11), there are some differences during mid-summer and spring. The summer values differ among sites with for instance a mean December value close to 15 ppbv at NM (16.2 ppbv) and HA (15.1 ppbv) compared to 20 ppbv at DDU. This difference is reduced when days with air transport to DDU from above 3000 m elevation are discarded (16.1 ppbv instead of 20 ppbv).

The same is observed in November with DDU mean value of 24.2 ppbv (22 ppbv without considering air masses coming from the high plateau) compared to 20.7 ppbv at NM. This suggests that the difference between DDU and the two western sites of NM and HA is due to katabatic conditions, as already discussed by Legrand et al. (2009). The case of SY appears to be intermediate with November and December values of 22.7 and 18.2 ppbv respectively.

The larger day-to-day summer variability in ozone at DDU compared to that at other coastal sites (Fig. 11) is also evidenced

by examining the frequency distribution of daily values observed in December at the 4 coastal sites (Fig. 12). While all sites display maximum occurrences of ozone mixing ratios between 15 and 19 ppbv, more high ozone values (17% above 25 ppbv, 33% above 22 ppbv) are observed at DDU compared to NM (1% above 25 ppbv, 5% above 22 ppbv) and HA (less than 1% above 25 ppbv, 1% above 22 ppbv). For the case of SY an intermediate situation is observed with 2.5% of values above 25 ppbv and 12% of values above 22 ppbv. That confirms that the rather high ozone levels observed at DDU in summer are

driven by the more frequent occurrence of katabatic winds bringing oxidant rich air masses from the Antarctic plateau toward the margin regions of East Antarctica.

In spring a particularly high day-to-day variability can be observed from August to October at the two western coastal sites of NM and HA (Fig. 11). As seen in Figure 13 this high variability of values is primarily due to occurrence of low ozone mixing ratios. Fewer low ozone values (less than 1% below 22 ppbv) are observed at DDU over these three months

compared to NM (11.4% below 22 ppbv) and HA (17.8% below 22 ppbv). Note that no significant difference was observed when daily values calculated without removing hourly values suspected to be contaminated (i.e. lower ozone values, see Sect. 2) were considered. Thus no bias due to inadequate removal of low values may have caused the observed absence of ozone values lower than 20 ppbv at DDU. At NM in spring Frieß et al. (2004) examined surface ozone data and differential optical absorption spectroscopy (DOAS) observations of BrO along with the time air masses were located close to the sea ice

surface as determined using back trajectory calculations. They showed that whereas ozone levels remain close to typical spring background level of 30 ppbv in air masses having been for a short contact with sea ice (less than 20 hours), severe ozone depletions are observed in air masses having been in contact with sea-ice for a long time (down to 5 ppbv of ozone for



air masses having sea-ice contact of 5 days). At DDU, with a sea ice extent offshore the site of around 450 km (i.e. more than 3 times reduced compared to NM, see Fig. 1) a long time contact of air masses with sea-ice is very rare. In October for instance, during only 5 occasions the air masses reaching DDU over the course of a whole day had travelled exclusively over the ocean during the 5 days prior their arrival. The time contact of air with sea-ice shown by backward trajectory was twice

of 10 hours, twice of 20 hours, and once of 60 hours.

For the case of SY an intermediate situation is observed with 6.6 % of values below 22 ppbv. These inter-site differences in spring clearly follow the sea-ice area located offshore the site with a minimum offshore DDU and a maximum offshore HA (see Fig. 1). It is interesting to emphasize that most values around 15 ppbv were encountered at HA in October at the time that surface BrO mixing ratios were found to reach a maximum (Saiz-Lopez et al., 2007).

**6. Intra-decadal variability of ozone**

The long-term ozone trends at the two East Antarctic sites were examined by calculating the regression line slopes through annual and/or monthly mean values. At Concordia, no annual analysis was done since a significant portion of summer measurements was missed (in 2007, 2008, 2009, 2011, and 2013). At DDU, a very weak annual increasing rate of $0.07 \pm 0.07$ ppbv yr$^{-1}$ is calculated, however the regression line slope was found to be not statistically different from zero at the $P >$

95% confidence level. The mean slopes of the linear regression lines calculated for each month are reported in Fig. 14 together with those calculated over the similar time period (from 2004/07 to 2014) at SP, SY, NM, and HA. It can be seen that, at DC and DDU, a decreasing trend is detected in summer (from November to February) and an increasing trend in winter.

In summer, all sites except HA for which the uncertainties of the regression line slopes are rather high, exhibit a decreasing

trend in summer but they become statistically different from zero at the $P > 95\%$ confidence level only in December at DDU and in January at the South Pole. At Concordia, as discussed in section 3.1.1, a large part of the inter-annual monthly variability in summer can be attributed to difference in synoptic transport of air masses over the Antarctic continent prior to arrival at the site. A good example of the role of transport is seen in November for which a relatively strong decreasing trend of 0.38 ppbv yr$^{-1}$ is calculated (Fig. 14). As seen in Table 1, the decrease in November over the 2007-2014 time period at

Concordia is mainly driven by the occurrence of the highest value (33.1 ppbv) encountered in 2007 (33.8 ppbv for the second half of November, Table 1) and of the lowest value (27.2 ppbv) in 2013 (28.6 ppbv for the second half of November, Table 1). At DDU, the decreasing trend detected in December is also partly due the fact that the two recent years experienced low values, in 2013 due to a particularly high fraction of marine air arriving (44% of time instead of $17 \pm 3$ % 3 over other years, not shown) and in 2012 to a low contribution of air masses coming from the high plateau (10% instead of $19 \pm 4$ %

over other years, not shown).

Though being often not significantly different from zero at the $P > 95\%$ confidence level, the recent decreasing trends (2004/07-2014) detected in summer differ from those reported by Helmig et al. (2007a) who showed significant increasing



trends in summer particularly at SP from 1975 to 2005. Similarly to Concordia, synoptic transport conditions were shown to influence surface ozone level at SP (Helmig et al., 2008). Therefore the trend reversal of surface ozone layer over the last decades at SP, i.e. an increase between 1975-2005 and a decreasing trend thereafter, may reflect a modification of synoptic transport favouring a larger impact of air masses coming from the highest part of the plateau from 1975 to 2005 and less

from 2005 to 2014. Another parameter that has be considered when discussing trends over the four last decades is the total ozone column whose variations may modify the amount of UV radiations reaching the ground and thus $NO_x$ snow emission rates and ozone photochemical productions. At the SP the total ozone column (TOC expressed in DU) in January decreased at a rate of 1.6 DU per year from 1975 to 2004 (TOC = -1.6213x + 307 with $R^2$ = 0.77) but has increased at a rate of 2.1 DU per year from 2004 to 2014 (TOC = 2.1126x + 262.2 with $R^2$ = 0.41). Though the link between the total ozone column and

surface ozone mixing ratios is complicated since other factors (snow nitrate concentration, atmospheric mixing, snow fall, etc.) influence snow $NO_x$ emission rates (Frey et al., 2015), we cannot totally rule out that in addition to synoptic transport conditions changing total ozone column may have contribute to change detected in surface ozone at the surface over the last decades. Note that the changes in the total ozone column at the SP in January are not related to an on-going recovery of the spring stratospheric layer since in October the total ozone column which have decreased at a rate of 4.2 DU per year from

1975 to 2004 (TOC = -4.2x + 251 with $R^2$ = 0.53) does not change significantly between 2004 and 2014 (TOC = 2.7x + 147 with $R^2$ = 0.098). Though the record at Concordia only covers the last decade, the total ozone column measured at Concordia by the SAOZ reported in Table 1 indicate no obvious relationship between stratospheric and surface ozone, the inter-annual variability of surface ozone levels being driven by synoptic transport as previously discussed in Sect. 3.1.1. Finally, the observation of a significant recent summer decreasing trend at DDU in December (Fig. 14) is consistent with the strong

influence at that site of processes acting inland Antarctica. On the contrary, the weaker decreasing trend in summer calculated for other coastal sites like NM and SY (Fig. 14) is also consistent with a weaker impact of oxidant-rich inland air masses at these two coastal sites as previously discussed in Sect. 5.
The increase of ozone during winter seen at all sites (except at HA) over the 2004/07-2014 time period is a quite interesting observation (Fig. 14). Whereas none of the increasing winter trends was found to be statistically different from zero at the *P*

> 95% confidence level at Concordia, at DDU increasing trends that range between 0.2 and 0.3 ppbv yr$^{-1}$ are statistically different from zero at the *P* > 95% confidence level from April to July. None of them are significantly different from zero at South Pole and Neumayer but interestingly, as for DDU, they are significant at Syowa. Given the lack of solar irradiance during the winter months in Antarctica, local photochemical processes are unlikely to have an influence. Therefore, it seems more probably that this ozone increase reflects changes in stratosphere-troposphere (or synoptic transport of ozone-enriched

air to these sites) and/or change in the stratospheric ozone reservoir. That motivates further works to extend the time series and examine the role of these parameters that are expected to control the budget of surface ozone at the surface in winter.





### 7. Concluding remarks

Recently obtained surface ozone measurements at the coastal site of Dumont d'Urville and the inland Concordia station allowed extending data from on-going measurements made at the South Pole and at several coastal sites located either in West Antarctica (Neumayer and Halley) or to East Antarctica (Syowa). Similar features in the occurrence of high ozone

events are observed at the two inland sites of Concordia and the South Pole in summer. However, near surface values exceed 40 ppbv at the South Pole but not at Concordia. Investigation of the vertical ozone distribution at the two sites indicates that the differences at the surface are related to boundary layer dynamics, in particular the presence of a diurnal radiation and temperature cycles at DC, which is lacking at SP, and associated cycles of boundary layer stability and mixing. We present indications for decreasing surface ozone trends in the summer at the DC and SP over the most recent observation period

(2004/07-2014). Though synoptic transport largely influences ozone level at the two sites, we cannot rule out that this change in surface ozone in summer is also related to an increase of the stratospheric ozone layer and associated reduction in $NO_x$ photochemical production from the snow, $NO_x$ emissions from the snowpack, and resulting ozone photochemical production. The unique character of DDU with respect to other coastal Antarctic sites is emphasized by its susceptibility to inland $NO_x$ chemistry, showing consistently a decreasing trend in summer over the 2004/07-2014 years in accordance with

the inland sites. On the contrary, the DDU ozone record shows a weaker signal of bromine chemistry that is evident at Halley and Neumayer in spring.




**Acknowledgements.**

The OPALE project was funded by the ANR (Agence National de Recherche) through contract ANR-09-BLAN- 0226. National financial support and field logistic supplies for the summer campaign were provided by the Institut Polaire Français-Paul Emile Victor (IPEV) through program n°414 and 903. This work was initiated in the framework of the French environmental observation service CESOA (Etude du cycle atmosphérique du Soufre en relation avec le climat aux moyennes et hautes latitudes Sud, http://www-lgge.obs.ujf-grenoble.fr/CESOA/spip.php?rubrique2) with the financial support of INSU (CNRS). Total ozone column data from the South Pole are available from Earth System Research Laboratory (Global monitoring division, http://www.esrl.noaa.gov/gmd/dv/data/) (Petropavlovskikh, I., Oltmans, S.J. (2012), Dobson Total Ozone measurements, Version: 2013-02-25, Path: ftp://aftp.cmdl.noaa.gov/data/ozone/in-situ/total_o3). Total ozone column at Concordia are from the SAOZ (Système d'Analyse par Observation Zénitale), which is part of NDACC (Network for the Detection of Atmospheric Composition Change). SAOZ measurements were made with the support of OVSQ (Observatoire de Versailles Saint-Quentin-en-Yvelines) and the IPEV program n°209. We thank Marie Dumont from Centre d'étude de la neige (CEN/Météo-France) and Guiseppe Camporeale from ENEA who contributed to balloon soundings done in summer 2009/2010. We also thank Christophe Genthon from LGGE who provided us the Vaisala PTU sondes in the frame of the Concordiasi program, Marion Marchand from LATMOS who provided us the ozone sondes and calibration system in the frame of the program IPEV n°912 during summer 2009/2010.

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



**Tables**

**Table 1.** Averaged ozone mixing ratios during second half of November, in December and during the first half of January along with the mean length of 5-day backward trajectories at Concordia and total ozone column measured by the SAOZ (see Sect. 6). Only time periods during which more than 80% of data were available were considered.

| Months | Year | O$_3$ (ppbv) | Trajectory Length (km) | Total Ozone Column (DU) |
|---|---|---|---|---|
| November (Second half) | 2007 | 33.8 | 2572 | 274 |
| | 2008 | 29.9 | 2928 | 245 |
| | 2009 | 30.4 | 3031 | 352 |
| | 2010 | 31.6 | 2400 | 220 |
| | 2011 | 33.5 | 2903 | 320 |
| | 2012 | 31.3 | 2860 | 346 |
| | 2013 | 28.6 | 3183 | 322 |
| December | 2009 | 29.6 | 2268 | 327 |
| | 2010 | 26.6 | 2852 | 272 |
| | 2011 | 29.3 | 2496 | 290 |
| | 2012 | 30.8 | 2434 | 336 |
| January (First half) | 2008 | 24.0 | 2257 | 316 |
| | 2010 | 27.5 | 2286 | 305 |
| | 2012 | 22.0 | 3196 | 304 |
| | 2013 | 24.9 | 2284 | 326 |
| | 2014 | 24.0 | 2690 | 306 |





**Table 2.** Chemical measurements available from Concordia and Dumont d'Urville gained during the OPALE campaigns (in 2010/2011 at DDU, in 2011/2012 at Concordia) that were used to calculate production and losses of ozone reported in Figure 10 (see Sect. 3.1.2). Values are 24-h averages. * Estimated values from $NO_2$ measurements.

| | Concordia | DDU | References |
|---|---|---|---|
| $RO_2$ (molecule cm$^{-3}$) | $10^8$ | $3.4 \ 10^8$ | Kukui et al. (2012), Kukui et al. (2014) |
| $HO_2$ (molecule cm$^{-3}$) | $7 \ 10^7$ | $1.8 \ 10^8$ | Kukui et al. (2012), Kukui et al. (2014) |
| OH (molecule cm$^{-3}$) | $3.1 \ 10^6$ | $2.1 \ 10^6$ | Kukui et al. (2012), Kukui et al. (2014) |
| NO (pptv) | 60 | 15* | Grilli et al. (2013), Frey et al. (2015) |
| $O_3$ (ppbv) | 24 | 21 | Preunkert et al. (2012), This work |





**Table 3.** Comparison of typical vertical gradient of ozone above the ground at Concordia in summer in the afternoon and those observed at the South Pole. For the South Pole we report the composite based on 88 profiles obtained during summer months between 1991 and 2005 showing vertical gradient of more than 8 ppbv (Oltmans et al., 2008) and the case of 24 December 2003 characterized by a mixing ratio close to 50 ppbv near the surface reported by Helmig et al. (2008).

| Site (launch time) | Ozone mixing ratio within the PBL | Ozone mixing ratio above the PBL |
|---|---|---|
| Concordia (29 December 2009 16:53) | 22.1 ± 0.15 ppbv (3215-3620 m asl) | 16. 5 ± 1.3 ppbv (3700-4140 m asl) |
| Concordia (31 December 2009 15:30) | 19.4 ± 0.14 ppbv (3210-3610 m asl) | 16. 6 ± 0.13 ppbv (3680-3800 m asl) |
| Concordia (4 January 2010 14:35) | 33.0 ± 0.5 ppbv (3215-3463 m asl) | 20. 8 ± 1.7 ppbv (3558-4225 m asl) |
| Concordia (11 December 2011 14:54) | 34.0 ± 1.1 ppbv (3215-3438 m asl) | 25. 1 ± 2.0 ppbv (3450-3970 m asl) |
| South Pole (24 December 2003) | 48.0 ± 0.5 ppbv (2850-2900 m asl) | 24.0 ppbv (3020-3350 m asl) |
| | 40.0 ppbv (2950 m asl) | |
| | 32.5 ppbv (3000 m asl) | |
| | 27.5 ppbv (3020 m asl) | |
| South Pole (88 profiles in summer) | 32.7 ± 0.2 ppbv (2850-2900 m asl) | 22.0 ppbv (3300-3800 m asl) |
| | 31.5 ppbv (2950 m asl) | |
| | 29.8 ppbv (3000 m asl) | |
| | 26.3 ppbv (3100 m asl) | |
| | 24.0 ppbv (3200 m asl) | |





**Figures**

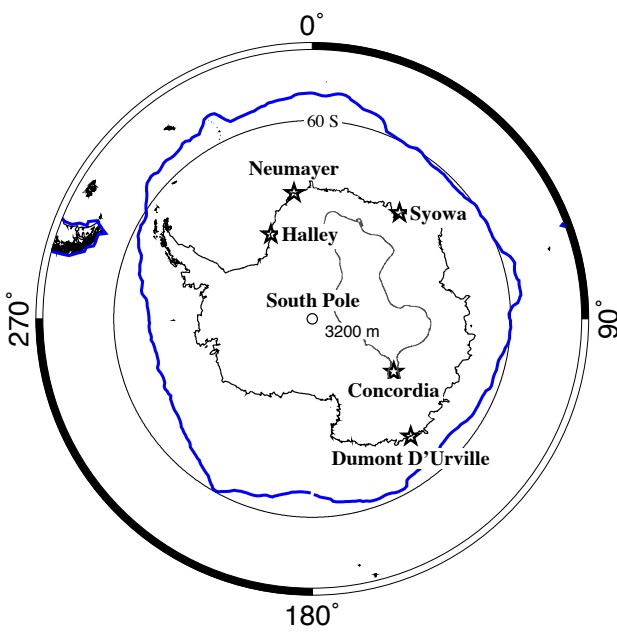

**Figure 1.** Map of Antarctica showing the inland sites of the South Pole and Concordia, the coastal sites of Dumont d'Urville and Syowa in East Antarctica, of Neumayer and Halley in West Antarctica. The blue line refers to the mean location of the
10  sea ice edge end of winter (August) over the period 1981–2012 (NOAA_OI_SST_V2 data provided by the NOAA/OAR/ESRL PSD, Boulder, Colorado, USA, http:// www.esrl.noaa.gov/psd).



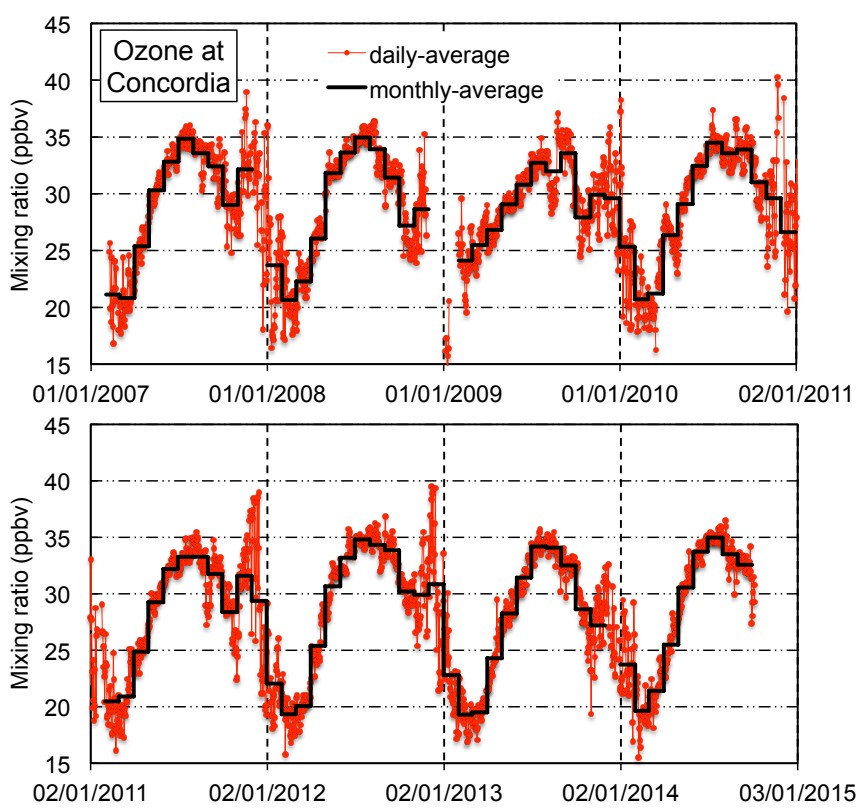

**Figure 2.** Daily averaged (red points) and monthly averaged (solid black line) surface ozone mixing ratio at Concordia from February 2007 to October 2014.





**Figure 3.** Daily averaged (red points) and monthly averaged (solid black line) surface ozone mixing ratio at Dumont

5   d'Urville from March 2004 to October 2014.





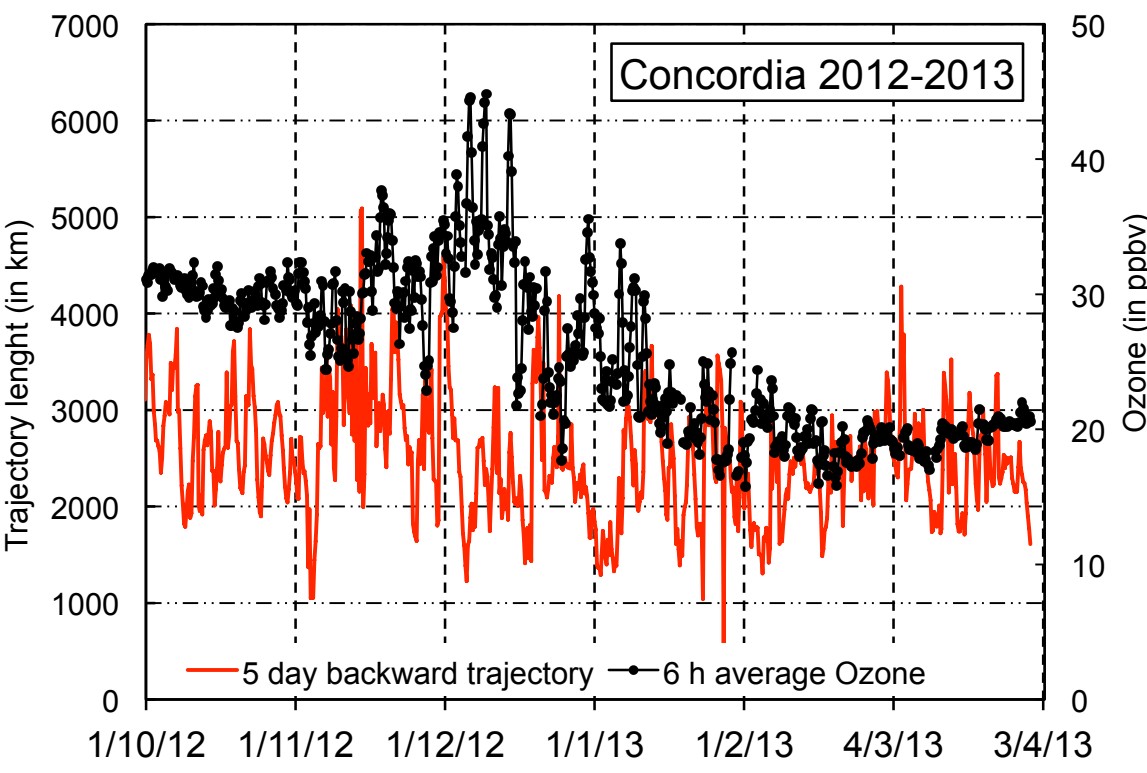

**Figure 4.** 6 h averaged ozone mixing ratio and distance covered by the corresponding 5 day backward trajectory at Concordia from spring 2012 to fall 2013.


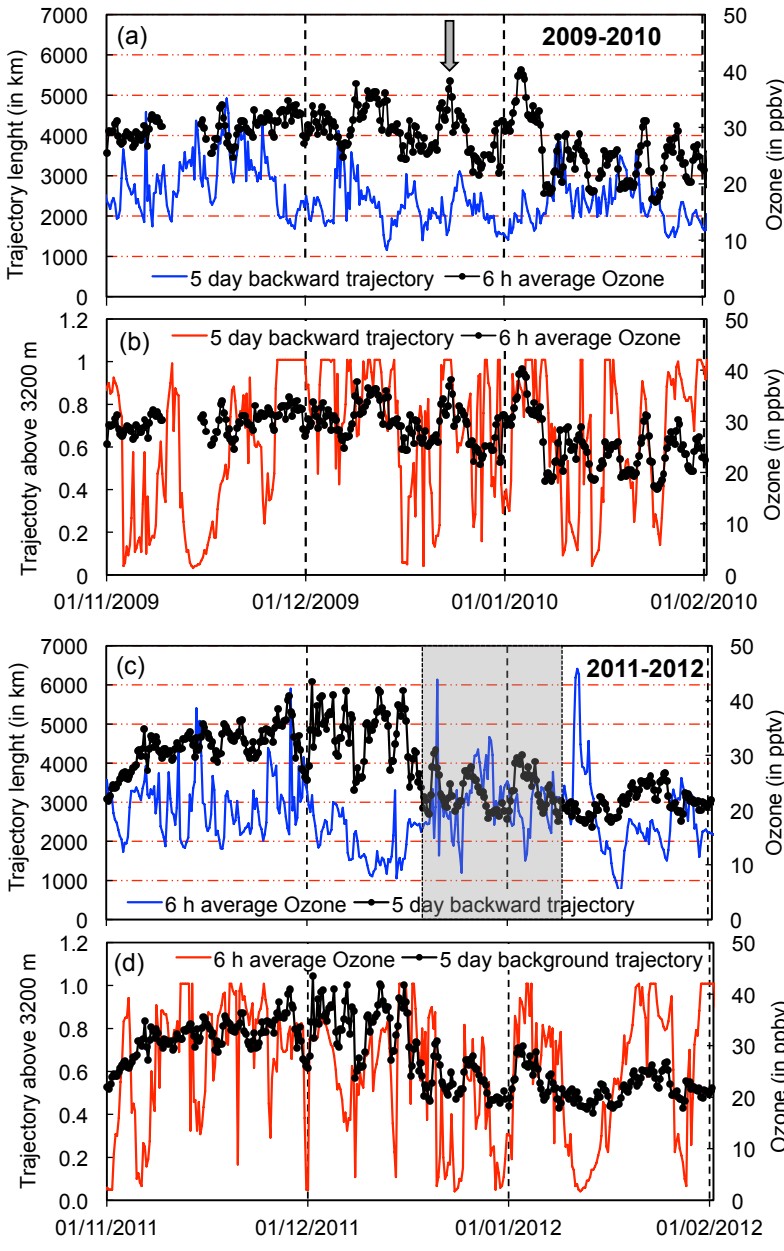

**Figure 5.** (a) and (b): 6 h averaged ozone mixing ratio (black dots) and the corresponding 5 day backward trajectory at Concordia from spring 2009 to fall 2010. The blue line is the trajectory length, the red line the fraction of the trajectory being above 3200 m elevation. The grey arrow refers to the day December 23 during which the observed fast increase of ozone is discussed in Sect. 3.1.1. (c) and (d): same as (a) and (b) for spring 2011 to fall 2012. The grey area denotes the sampling time period of the OPALE campaign.





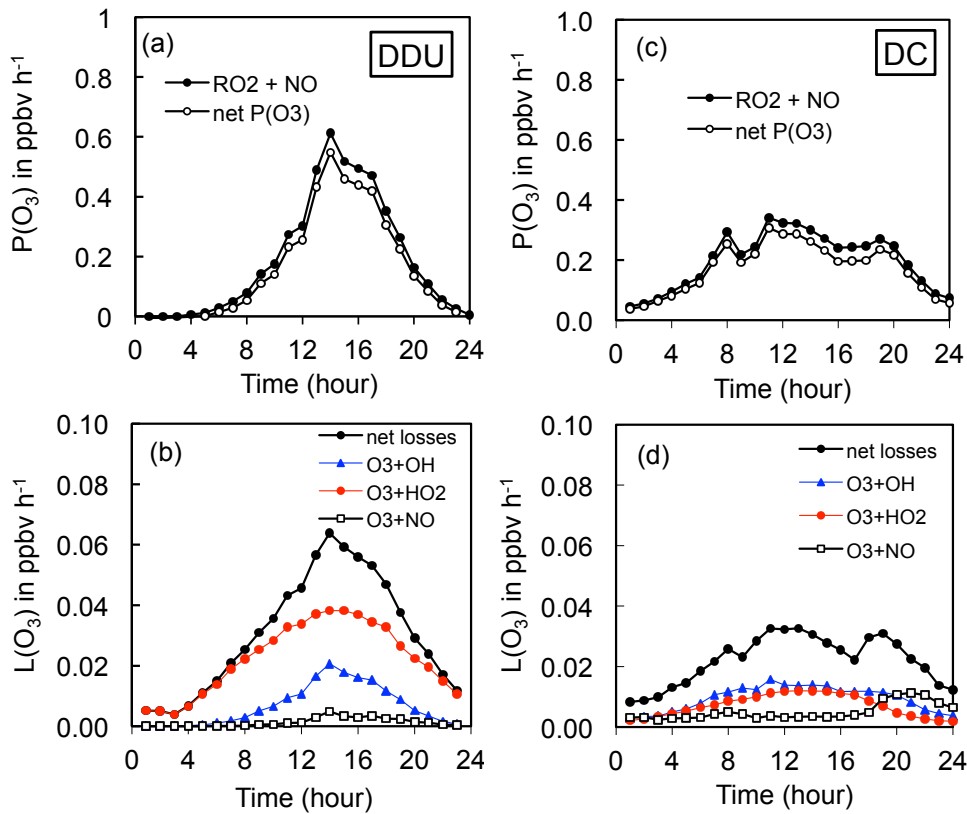

**Figure 6.** Ozone productions (a and c) and losses (b and d) at DDU and DC calculated on the basis of observations summarized in Table 2.



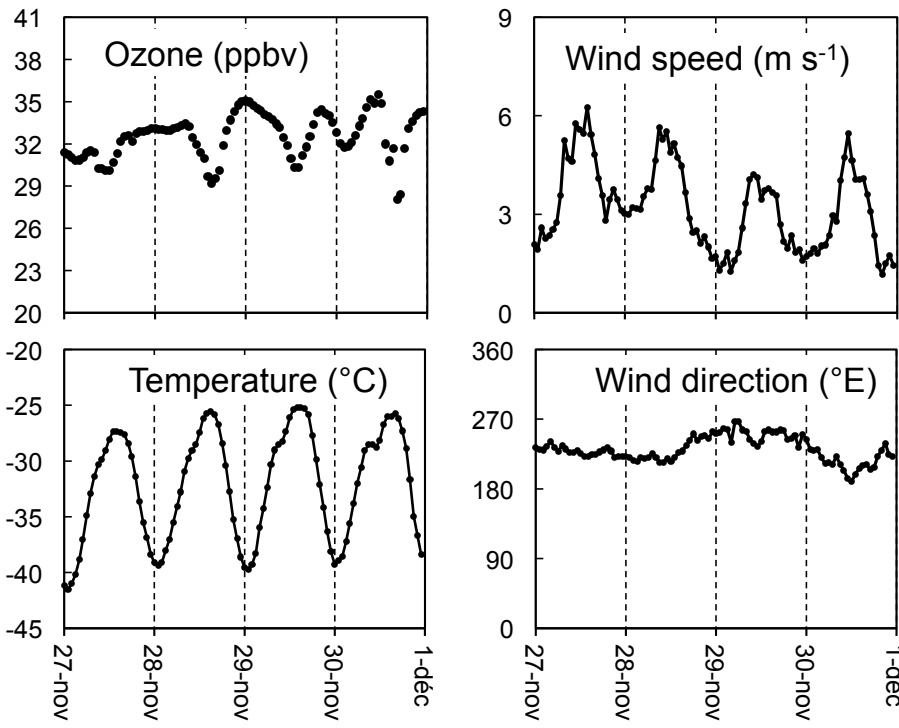

**Figure 7.** Hourly averaged surface ozone mixing ratio together with weather conditions at Concordia in summer over four
10    days (27 to 30 November 2009) characterized by stable continental conditions (see also Fig. 5a).





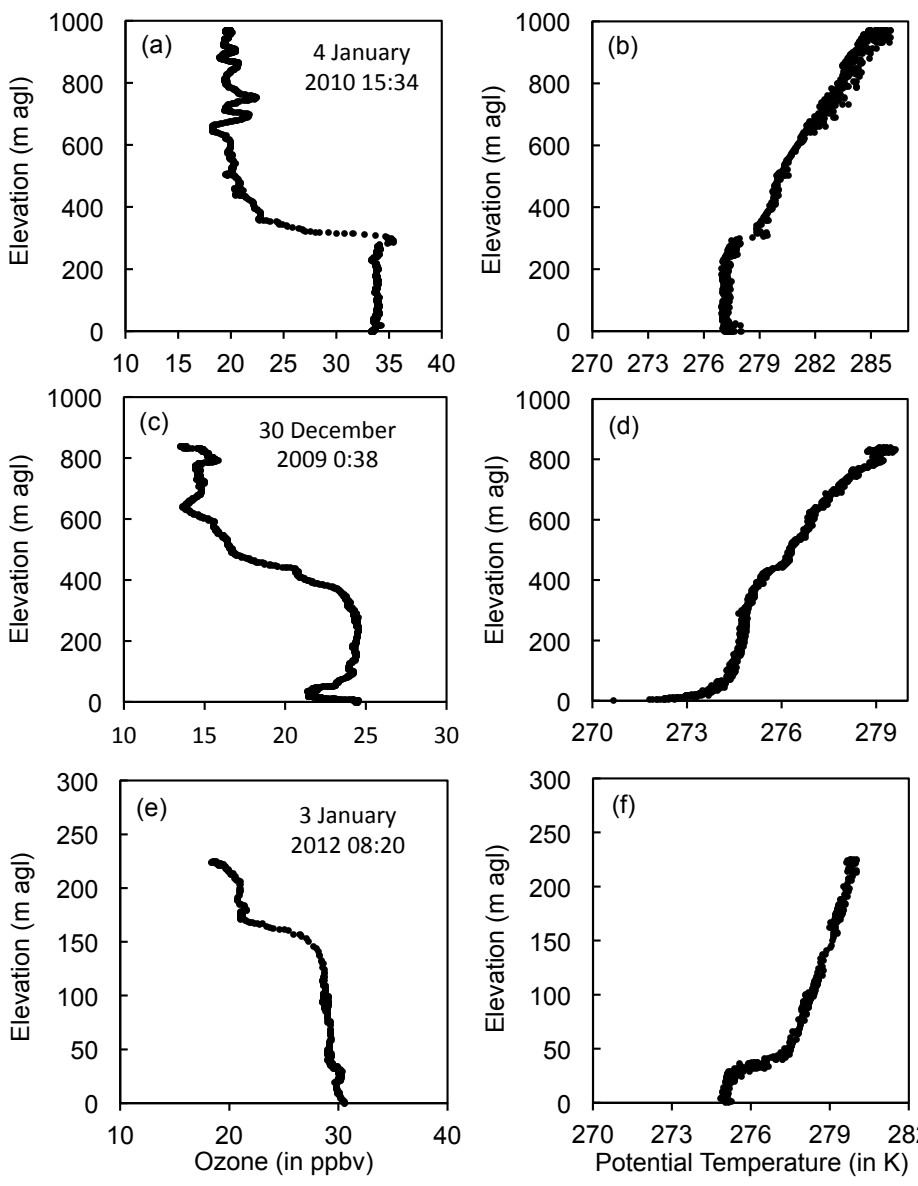

**Figure 8.** Examples of vertical distributions of ozone at Concordia in the afternoon (a), around mid-night (c), and early morning (e) along with the corresponding potential temperature profiles (b, d, and f, respectively). Dates are reported in local time and elevation in meters above ground level (agl).



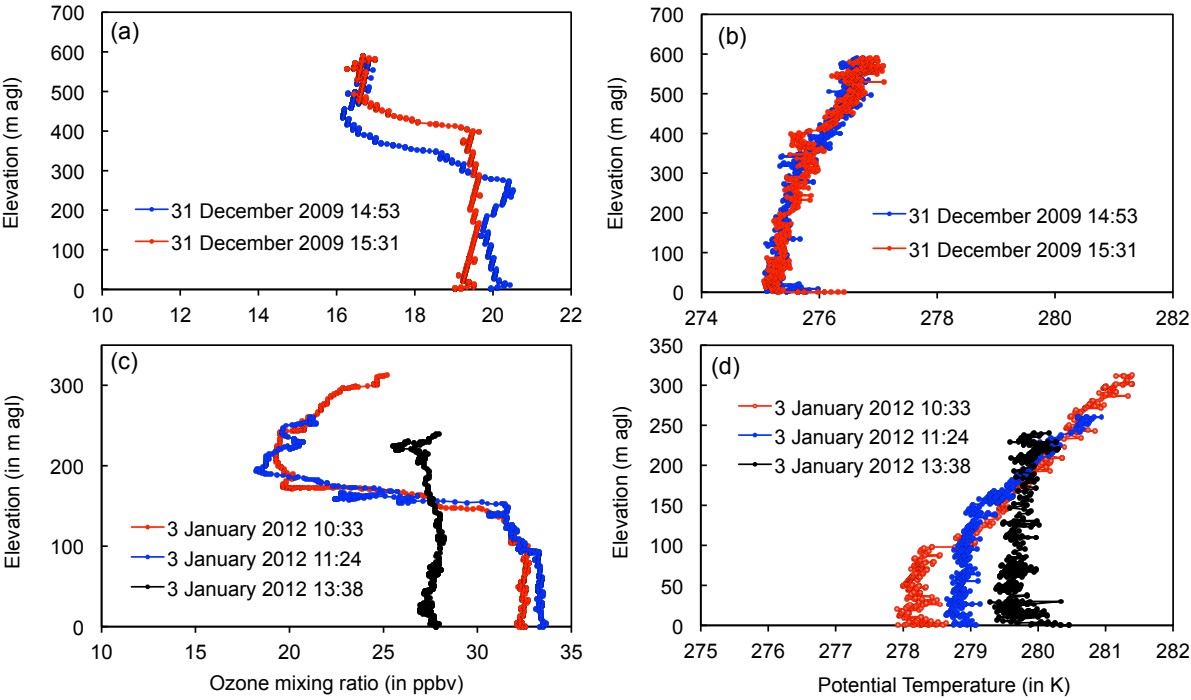

**Figure 9.** Examples of vertical ozone distribution changes at Concordia (during the afternoon (a), and from the morning to the afternoon (c)), along with corresponding potential temperature profiles (b and d, respectively). Dates are reported in local time and elevation in meters above ground level (agl).





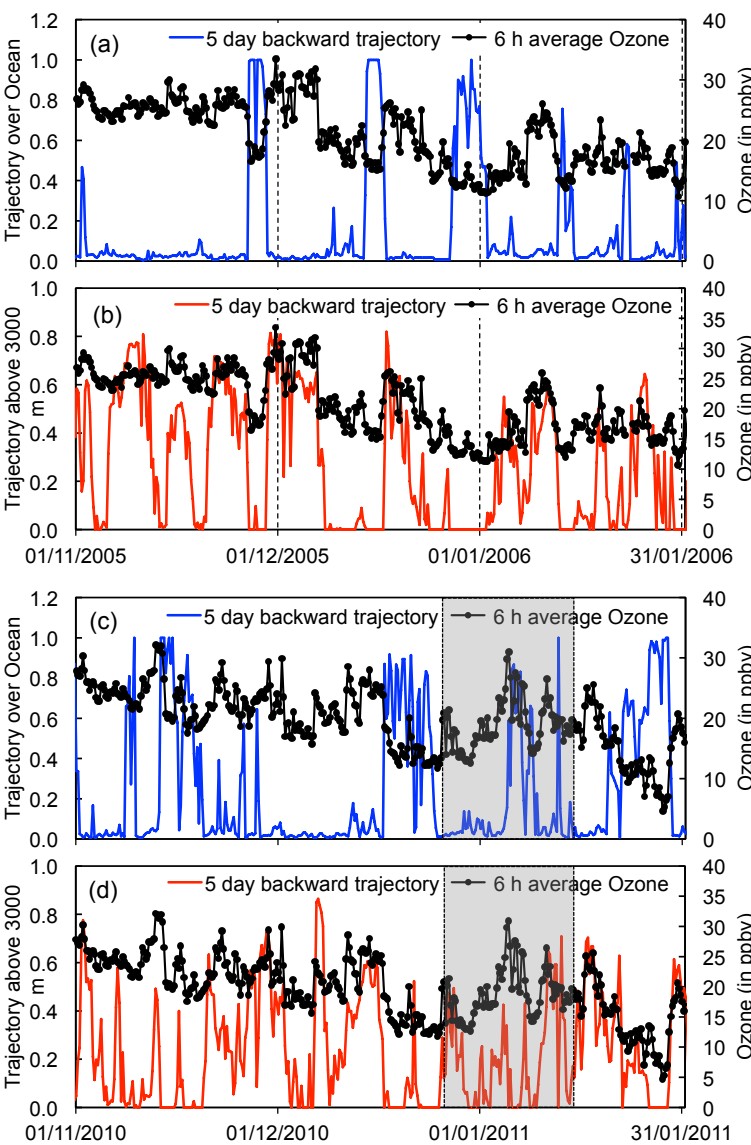

**Figure 10.** (a) and (b): 6 h averaged ozone mixing ratio (black dots) and the corresponding 5 day backward trajectory at DDU from spring 2005 to fall 2006. The blue line is the trajectory length, the red line the fraction of the trajectory being above 3200 m elevation. The grey arrow refers to the day December 23 during which the observed fast increase of ozone is discussed in Sect. 4. (c) and (d): same as (a) and (b) for spring 2011 to fall 2012. The grey area denotes the sampling time period of the OPALE campaign.





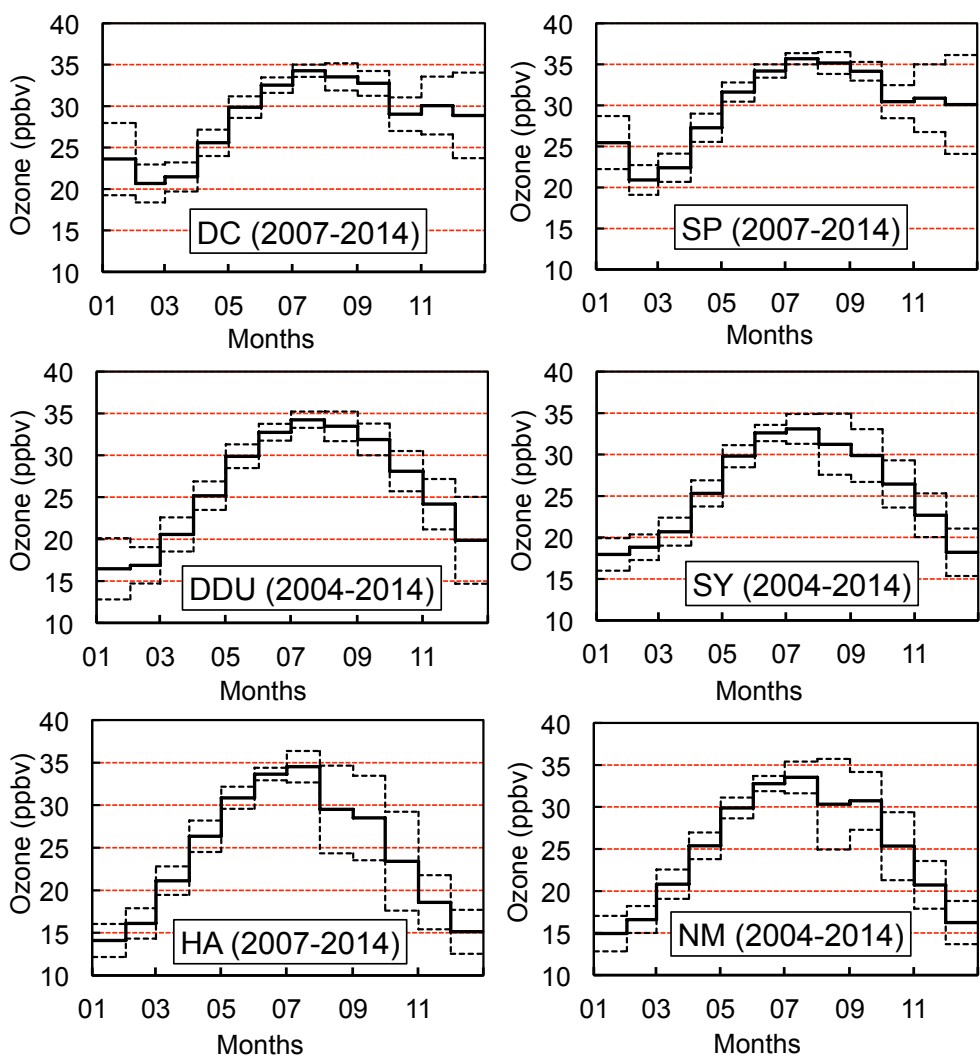

**Figure 11.** Monthly averaged ratios and standard deviations of ozone mixing ratios at Concordia (DC), South Pole (SP),
Dumont d'Urville (DDU), Syowa (SY), Halley (HA), and Neumayer (NM). The standard deviations represent day-to-day
variability.





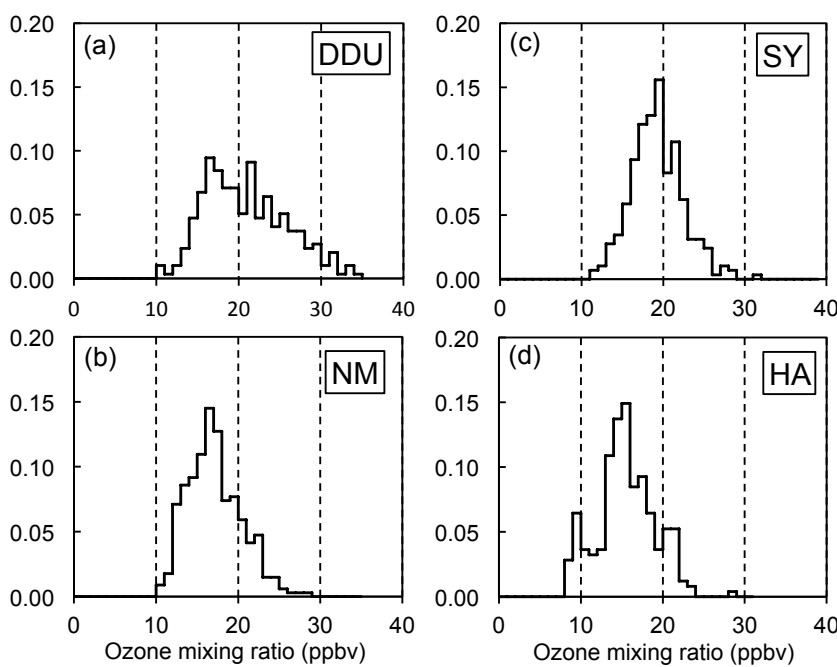

**Figure 12.** Frequency distribution of daily averaged $O_3$ mixing ratios observed in December (2004-2014) at the coastal sites of Dumont d'Urville (a), Neumayer (b), Syowa (c), and Halley (d).





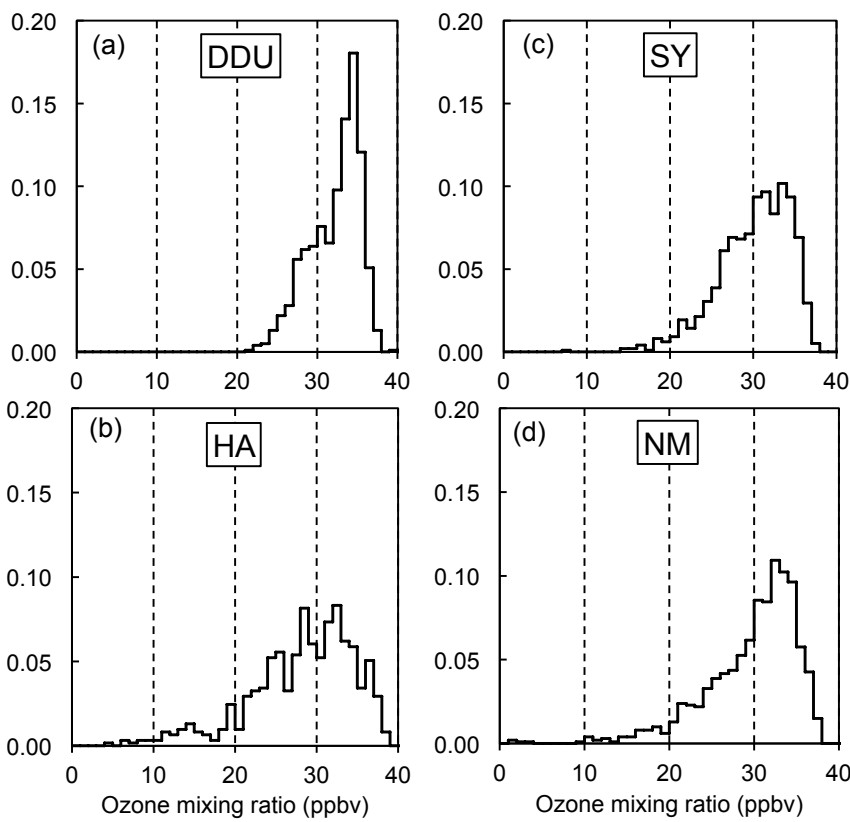

**Figure 13.** Frequency distribution of daily averaged ozone mixing ratios of O₃ observed in August, September, and October (2004-2014) the coastal sites of Dumont d'Urville (a), Halley (b), Syowa (c), and Neumayer (d).



**Figure 14.** Monthly slopes of the linear regression lines of ozone trends analyses (in ppbv yr$^{-1}$) at Concordia (DC), South Pole (SP), Dumont d'Urville (DDU), Syowa (SY), Halley (HA), and Neumayer (NM). Black circles denote slope values that are statistically different from zero at $P > 95\%$.