# Peer review of "Inter-annual variability of surface ozone at coastal (Dumont d'Urville, 2004-2014) and inland (Concordia, 2007-2014) sites in East Antarctica"

_Atmospheric Chemistry and Physics, 2016_

## Referee Comment (RC1) · Anonymous Referee #3 · 23 Mar 2016

General Comments:

This manuscript provides a nice synopsis of multi-year surface ozone records at two Antarctic locations: almost 8 years at the inland Concordia Station, and almost 11 years at the coastal Dumont d'Urville Station. A comparative analysis is presented between these stations, South Pole, Syowa, Neumayer and Halley, and the role of various factors influencing the characteristics of the annual and diurnal cycles of surface ozone, such as topography, meteorology, proximity to the ocean, and oxidant chemistry, is examined. With several relatively minor adjustments, this manuscript will be appropriate for publication in ACP.

[Figure]

The most significant issue I would like to see addressed is analysis presented in section 3.1.1. on the relationship between trajectories and ozone concentrations. This is rather unsatisfying, since it is based on a "pick and choose" approach instead of the analysis of patterns that can be rigorously generalized. It also places more trust in trajectory models than they probably deserve, particularly beyond 4-5 days out. I took the liberty of plotting the data in the first two columns of Table 1 (see attached figure). There is no need for me to expand on this plot, since it says everything that needs to be said. It is of course true that there appear to be some correlations in Fig. 4 if some periods are inspected in isolation, but this method is scientifically not terribly convincing. An analysis along the lines of what was presented in Bottenheim and Chan (2006) may, at least partially, alleviate these concerns. With 8 and 11 years of data, there should be enough material here that this type of analysis would provide further insights.

Specific Comments:

Page 1 Line 24: cycles
P1L26: but not at the South Pole, and a ... This sentence needs to be rephrased. Do you mean "far better mixed layer with higher ozone concentrations"? Also, the 250m needs to be reconciled with numbers in section 4.
P2L5: research; separate the effect
P2L15: remove "that"
P2L22: (Crawford...
P3L18: papers in
P3L32: at DDU
P4L1: were made once a month
P4L21: treat McClure et al like any other reference: just give the authors and date, the rest goes to the end of the paper
P4L28: prior to
P4L31: the upward shift: if the delay time is predictable, then the shift can be corrected.

Was this done to finalize the data?

P4L33: This permitted the documentation of the ...

P5L2: proved

Section 2 should also briefly explain SAOZ (which is used in Table 1 but never spelled out until the Acknowledgements). And it needs to include some details on the trajectory model used. These models are not absolute, and some are better than others. 5 days out, most of them have rather large uncertainties.

I think p.6 can be shortened to one paragraph, and table 1 should be eliminated, because it does not provide convincing evidence of any patterns, even if the patterns the authors strain to see make physical sense.

P7L10: invariant instead of "not modified".

P7L26: remove "neutral"; convective conditions would conceivably also produce a straight profile

P8L1: no longer

P8L2: generally several hundreds of m deep as seen...

P8L7: radiation

P8L32: Details

P9L6: remove "very"

P9L17: dependent

P8L23: lasted to the

P9L30: remove "over"

P10L19: absence of detectable increase: what about the potential role of meteorology / mixing? Also, I suspect the diurnal cycle of O3 at Concordia isn't always as nice as that shown in Fig. 7.

P11L5: radiation... acts

P11L9: dynamics

P11L27: the "always" seems to contradict the statement in L19 about 25% of profiles not having significant vertical gradients

P13L5: remove the "of"s

P14L20: On the other hand,

P15L7: of diurnal

P15L8: which are

P15l15: On the other hand

The Conclusions should also mention the winter trend. This is interesting as well.

Table 1: should be removed (see comments above). If you really wish to keep it, you need to discuss the SAOZ column somewhere in the text (which is currently not the case), and explain the colour code (red/blue/black numbers).

Table 2: use standard exponential notation

Table 3: give the heights in "agl" please, so they can be related to what's in the text

Fig. 4: identify the interesting episodes discussed in the text with arrows

Fig. 10: a grey arrow is mentioned in the caption, but I can' find it

Fig. 14: give a definition of the error bars in the caption

Reference:

Bottenheim, J. W. and Chan, E.: A trajectory study into the origin of spring time Arctic boundary layer ozone depletion, J. Geophys. Res., 111, D19301, doi:10.1029/2006JD007055, 2006.

[Figure]

$$O3 = 6E\text{-}05x + 28.45ppb$$
$$r^2 = 3E\text{-}05$$

**Fig. 1.** Concordia ozone - trajectory relationship

---

## Referee Comment (RC2) · Anonymous Referee #2 · 26 Apr 2016

General: This is a reasonably constructed paper largely focused on comparison of ozone measurements over the high plateau versus those in coastal areas over a period of less than a decade. A significant amount of text addressed ozone production versus loss terms in the context of several case studies from balloon profiling. Section 5, describing observations at other sites on the coasts, could be in Supplemental material so that the paper stays focused on high plateau (DC and SP) and provides a comparison with a coastal site (DDU). In a number of cases, it was hard to identify the "bottom line" or how robust the conclusions were. In terms of trends, in the face of decadal variability in the circulation around Antarctica, it would be useful to show some composite synoptic maps (e.g. high and low ozone at DDU) and assess if the

patterns have changed. In some of the discussion, it was easy to get lost in the details. It would also be nice to have some plots from ERA-I showing synoptic patterns rather than abstract statements of trajectories originating from x, y, or z (I looked at vector wind plots and maps of GPH for some of the examples.)

Specific areas for improvement: P4, L9 and L12: "Adjustment" or "change" are probably better words than "correction." P6, L7-20: The trajectory length analysis should be supported by a cluster plot showing the actual trajectories (or trajectory origins) overlain on the Antarctic region. This might help identify preferred source areas, if any. For example, do the trajectories follow terrain isopleths to DC or do they originate over the high terrain around Dome A? This could be a supplemental figure. P6, L21-23: This deserves some discussion of the high NOx fluxes in early December reported by Frey et al and the high surface nitrate at that time vis-à-vis the NOx-to-O3 conversion process. It is also unfortunate that the Gallee et al. modeling work for OPALE did not include O3 calculations. P6, L26+: Table I displays total column ozone with no discussion of its importance. Also there needs to be a few words explaining the red highlighted O3 values (the highest values in each time period.) P6: Was there any evidence of stratospheric intrusions. Although rare, Crawford et al 2001 showed one case in early December with an increase in 7Be while Traversi et al 2014 (Tellus Series B-Chemical and Physical Meteorology) argued for such a signature in nitrate measurements at Concordia.

P7, L10: Fig. 5a only shows length of trajectory not its origin. The claim of origin on the "high plateau" needs to be supported, especially where on the high plateau.

Section 3.1.2: The discussion of cases examining the competing effects of ozone production versus dilution in a growing convective boundary took a bit to follow before a final conclusion that dilution dominated the afternoon drop of ozone. The amount of data does not justify any robust conclusions. This should be emphasized. The authors should also look at the changing meteorology 1-4 January 2010 as a strong ridge developed over DC by 3 January (I looked at ERA-I results). The associated subsidence

might account for the shallow mixing layer on 3 January compared to 31 December.

P11, L9: "...difference in the dynamic of the lower atmosphere..." is vague.

P13, L4-5: This is confusing: "The time contact of air with sea-ice shown by backward trajectory was twice of 10 hours, twice of 20 hours, and once of 60 hours"

Section 5: This section should discuss the differences in the area of the topographic features that channel air from the interior to the coastal areas. Each of the stations may have distinctly difference source regions associated with local topography.

P12, L12,L20: "downslope" is more general. Low pressure systems on the coast can produce similar downslope conditions. It would be interesting to do a composite synoptic map for high and low ozone conditions at DDU etc.

Section 6. This section is rather speculative and could be tightened up considerably.

---

## Author Comment (AC1) · 26 May 2016

REV3: General Comments: This manuscript provides a nice synopsis of multi-year surface ozone records at two Antarctic locations: almost 8 years at the inland Concordia Station, and almost 11 years at the coastal Dumont d'Urville Station. A comparative analysis is presented between these stations, South Pole, Syowa, Neumayer and Halley, and the role of various factors influencing the characteristics of the annual and diurnal cycles of surface ozone, such as topography, meteorology, proximity to the ocean, and oxidant chemistry, is examined. With several relatively minor adjustments, this manuscript will be appropriate for publication in ACP. The most significant issue I would like to see addressed is analysis presented in section 3.1.1. on the relationship

between trajectories and ozone concentrations. This is rather unsatisfying, since it is based on a "pick and choose" approach instead of the analysis of patterns that can be rigorously generalized. It also places more trust in trajectory models than they probably deserve, particularly beyond 4-5 days out. I took the liberty of plotting the data in the first two columns of Table 1 (see attached figure). There is no need for me to expand on this plot, since it says everything that needs to be said. It is of course true that there appear to be some correlations in Fig. 4 if some periods are inspected in isolation, but this method is scientifically not terribly convincing.

AUT:Thank you for this comment: indeed Table 1 was not very convincing since everything was smoothed when averaged over one month or half a month. So we remove it and instead we simply plotted all backward trajectories for December (over the 7-8 years of the record) as a function of the corresponding ozone range. This figure far better illustrates the importance of the transport.

REV3:An analysis along the lines of what was presented in Bottenheim and Chan (2006) may, at least partially, alleviate these concerns. With 8 and 11 years of data, there should be enough material here that this type of analysis would provide further insights.

AUT:That is indeed another possibility but we think that this would be less straight-forward in Antarctica than in the case of Arctic. Anyway with the figure of backward trajectories introduced in the new version, the role of transport is clearly seen.

REV3:Specific Comments: Page 1 Line 24: cycles: Done P1L26: but not at the South Pole, and a ... This sentence needs to be rephrased. Do you mean "far better mixed layer with higher ozone concentrations"? Also, the 250m needs to be reconciled with numbers in section 4.

AUT:This long sentence was reworded into two sentences as follows: "However, in summer the diurnal cycle of ozone is different at the two sites with a drop of ozone in the afternoon at Concordia but not at the South Pole. The vertical distribution of ozone

above the snow surface also differs. When present, the ozone rich layer located near the ground is better mixed and deeper at Concordia (up to 400 m) than at the South Pole during sunlight hours."

REV3: P2L5: research; separate the effect AUT: Done

REV3: P2L15: remove "that" AUT: Done

REV3: P2L22: (Crawford... AUT: Done

REV3: P3L18: papers in AUT: Done

REV3: P3L32: at DDU AUT: Done

REV3: P4L1: were made once a month AUT: Done

REV3: P4L21: treat McClure et al like any other reference: just give the authors and date, the rest goes to the end of the paper AUT: Done

REV3: P4L28: prior to AUT: Done

REV3: P4L31: the upward shift: if the delay time is predictable, then the shift can be corrected. Was this done to finalize the data?

AUT: As mentioned in the text the shift is variable from one flight to another flight but never exceeded 20 m. The effect remains weak for the cases discussed in section 3.1.2.

REV3: P4L33: This permitted the documentation of the ... P5L2: proved AUT: Done

REV3: Section 2 should also briefly explain SAOZ (which is used in Table 1 but never spelled out until the Acknowledgements).

AUT: OK, since Table 1 was removed following the major issue rise at the beginning of your review, and following the recommendation of the other reviewer to reduce the discussion in Section 6 (restricting the discussion on summer trends to the South Pole) the SAOZ data at Concordia are not anymore needed.

REV3: And it needs to include some details on the trajectory model used. These models are not absolute, and some are better than others. 5 days out, most of them have rather large uncertainties.

AUT: Right and we add a paragraph at the end of section 2 as follows: "To characterize the origin of air masses reaching the DDU and Concordia regions, five day backward trajectories were computed using the Hybrid Single-Particle Lagrangian Integrated Trajectory model (R. R. Draxler and G. D. Rolph, NOAA Air Resources Laboratory, Silver Spring, Maryland, 2003, available at http://www. arl.noaa.gov/ready/hysplit4.html). Meteorological data from Global Data Assimilation Process (available at ftp://arlftp.arlhq.noaa.gov/pub/archives/gdas1) were used as input, and the model was run every 6 h in backward mode for three different altitudes (0, 250, and 500 m agl)."

REV3:I think p.6 can be shortened to one paragraph, and table 1 should be eliminated, because it does not provide convincing evidence of any patterns, even if the patterns the authors strain to see make physical sense.

AUT: Thank you for this comment. We agree and discarded this Table (including the SAOZ values now not discussed in section 6). Instead, we report traces of backward trajectories for the month of December (2007 to 2014) (a new figure, Fig. 6 has been introduced) to generalize the role of the transport on ozone values as also shown in the Figure 4 and 5 (but for specific time periods). The text was updated as follows: "Even on the monthly scale, during the summer the influence of the air mass trajectory on ozone levels is still visible. For instance, as seen in Fig. 6 for December, the highest ozone values (higher than 38 ppbv) correspond to the shortest trajectories that almost always spent a significant time fraction over the highest part of the plateau (above 3500 m asl). Trajectories corresponding to ozone values ranging between 32 and 38 ppbv travelled mostly above 2500 m asl in the inner part of the continent. Finally, ozone values below 26 ppbv are observed when the trajectories were among the longest, sometimes starting from coastal regions or from the ocean (Fig. 6)."

REV3: P7L10: invariant instead of "not modified". AUT: Done

REV3: P7L26: remove "neutral"; convective conditions would conceivably also produce a straight profile Right, AUT: Done

REV3: P8L1: no longer AUT: Done

REV3: P8L2: generally several hundreds of m deep as seen... AUT: Done

REV3: P8L7: radiation AUT: Done

REV3: P8L32: Details AUT: Done

REV3: P9L6: remove "very" AUT: Done

REV3: P9L17: dependent AUT: Done

REV3: P8L23: lasted to the AUT: Done

REV3: P9L30: remove "over" AUT: Done

REV3: P10L19: absence of detectable increase: what about the potential role of meteorology / mixing?

AUT: Yes that is possible but we cannot address this point (as we did for Concordia) since deploying tethered balloons is far more complicated at DDU compared to Concordia (more wind and turbulent regime).

REV3: Also, I suspect the diurnal cycle of O3 at Concordia isn't always as nice as that shown in Fig. 7.

AUT: You are right and this was explained at the beginning of section 3.1.2: only under very specific conditions is a diurnal cycle: "Given the importance of synoptic transport conditions on the day-to-day ozone variability (Sect. 3.1.1), the detection of an ozone trend over several days to estimate a mean local ozone production rate requires the search for a period of several days over which the transport pattern was invariant. As seen in Fig. 5a, from 27 to 30 November 2009 backward trajectories indicate transport

of air masses consistently from the high Antarctic plateau (100% of time above 3200 m asl), and over these 4 days surface wind direction remained unchanged (Fig. 8).

REV3: P11L5: radiation... acts AUT: Done

REV3: P11L9: dynamics AUT: Done

REV3: P11L27: the "always" seems to contradict the statement in L19 about 25% of profiles not having significant vertical gradients;

AUT: No since in fact here we are discussing the cases (75%) when a vertical gradient is present.

REV3: P13L5: remove the "of"s AUT: Done

REV3: P14L20: On the other hand, AUT: Done

REV3: P15L7: of diurnal, AUT: Done

REV3: P15L8: which are, AUT: Done

REV3: P15l15: On the other hand, AUT: Done

REV3: The Conclusions should also mention the winter trend. This is interesting as well.

AUT: Right and we added "Finally, the DDU data indicate a significant increasing trend in winter, motivating the extension of time series in view to examine the possible influence of stratosphere-troposphere transport for instance."

REV3: Table 1: should be removed (see comments above). If you really wish to keep it, you need to discuss the SAOZ column somewhere in the text (which is currently not the case), and explain the colour code (red/blue/black numbers).

AUT: As explained above we follow your suggestion to remove this Table. We also discard from the discussion the SAOZ data to shorten the discussions in Sect. 6.

[Figure]

REV3: Table 2: use standard exponential notation: AUT: OK Done

REV3: Table 3: give the heights in "agl" please, so they can be related to what's in the text: AUT: OK Done

REV3: Fig. 4: identify the interesting episodes discussed in the text with arrows: AUT: Done

REV3: Fig. 10: a grey arrow is mentioned in the caption, but I can' find it.

AUT: Thank you for this remark, in fact the caption was wrong. We now add arrows to highlight episodes discussed in the text.

REV3: Fig. 14: give a definition of the error bars in the caption:

AUT: Done, Vertical bars refer to standard errors of the regression lines.

---

## Author Comment (AC2) · 26 May 2016

REV2: General: This is a reasonably constructed paper largely focused on comparison of ozone measurements over the high plateau versus those in coastal areas over a period of less than a decade. A significant amount of text addressed ozone production versus loss terms in the context of several case studies from balloon profiling. Section 5, describing observations at other sites on the coasts, could be in Supplemental material so that the paper stays focused on high plateau (DC and SP) and provides a comparison with a coastal site (DDU).

AUT: Well, as stated in the introduction, the paper aims to fill the lack of ozone data in East Antarctica (both central and coastal). Also, the OPALE project addressed the

topics of oxidants at the central and coastal sites in East Antarctica. The comparison of the different coastal sites is also of great interest and thus we think that this coastal aspect is not at all marginal in the paper.

REV2: In a number of cases, it was hard to identify the "bottom line" or how robust the conclusions were. In terms of trends, in the face of decadal variability in the circulation around Antarctica, it would be useful to show some composite synoptic maps (e.g. high and low ozone at DDU) and assess if the patterns have changed. In some of the discussion, it was easy to get lost in the details. It would also be nice to have some plots from ERA-I showing synoptic patterns rather than abstract statements of trajectories originating from x, y, or z (I looked at vector wind plots and maps of GPH for some of the examples.).

AUT: OK in the revised version we report backward trajectories in summer (high and low ozone values) both for DDU and Concordia to better highlight the role of the transport on ozone levels (in addition to fig. 5 and 10).

REV2: Specific areas for improvement: P4, L9 and L12: "Adjustment" or "change" are probably better words than "correction."

AUT: We think that correction is correct in this context.

REV2: P6, L7-20: The trajectory length analysis should be supported by a cluster plot showing the actual trajectories (or trajectory origins) overlain on the Antarctic region. This might help identify preferred source areas, if any. For example, do the trajectories follow terrain isopleths to DC or do they originate over the high terrain around Dome A? This could be a supplemental figure.

AUT: As above mentioned we introduced a figure showing traces of backward trajectories from which region air masses reaching Concordia are coming from.

REV2: P6, L21-23: This deserves some discussion of the high NOx fluxes in early December reported by Frey et al and the high surface nitrate at that time vis-à-vis the

NOx-to-O3 conversion process. It is also unfortunate that the Gallee et al. modeling work for OPALE did not include O3 calculations.

AUT: We agree

REV2: P6, L26+: Table I displays total column ozone with no discussion of its importance. Also there needs to be a few words explaining the red highlighted O3 values (the highest values in each time period.

AUT: This Table (including the SAOZ values now not discussed in section 6, see below) was removed. Instead we report backward trajectories for the month of December (2007 to 2014) (a new figure, Fig. 6 has been introduced) to generalize the role of the transport on ozone values as shown for the Figure 4 and 5.

REV2: P6: Was there any evidence of stratospheric intrusions. Although rare, Crawford et al 2001 showed one case in early December with an increase in 7Be while Traversi et al 2014 (Tellus Series B-Chemical and Physical Meteorology) argued for such a signature in nitrate measurements at Concordia.

AUT: At least for summer, vertical ozone soundings from either the South Pole or Concordia do not reveal an importance of stratospheric intrusion for the surface ozone budget. We think that arguments developed by Traversi on nitrate are not directly helpful for the ozone concern (ozone is gaseous species, nitrate coming from the stratospheric reservoir would partly be due to PSCs sedimentation, Be is stuck on sub-micronic sulphate, ….).

REV2: P7, L10: Fig. 5a only shows length of trajectory not its origin. The claim of origin on the "high plateau" needs to be supported, especially where on the high plateau.

AUT: Right and in the new version we removed Table 1 that was not very useful and instead we plotted all backward trajectories for December (over the 7-8 years of the record) as a function of the corresponding ozone range. This figure far better illustrates the importance of the transport.

REV2: Section 3.1.2: The discussion of cases examining the competing effects of ozone production versus dilution in a growing convective boundary took a bit to follow before a final conclusion that dilution dominated the afternoon drop of ozone. The amount of data does not justify any robust conclusions. This should be emphasized.

AUT: We agree and reworded the text as follows: Though further investigations with diurnal vertical profiling over the course of the day are clearly needed, this example suggests that the decrease of ozone mixing ratio often detected near the surface in the afternoon (Fig. 8) may be related to dilution from the increase in the boundary layer depth which counteracts the local ozone photochemical production."

REV2: The authors should also look at the changing meteorology 1-4 January 2010 as a strong ridge developed over DC by 3 January (I looked at ERA-I results). The associated subsidence might account for the shallow mixing layer on 3 January compared to 31 December.

AUT: We agree but it should be at the border of the scope of this paper to describe synoptic conditions associated with day to day change of the vertical profiles.

REV2: P11, L9: ". . .difference in the dynamic of the lower atmosphere. . ." is vague.

AUT: Right, we added in parenthesis (more stratified lower atmospheric layers at the South Pole than at Concordia).

REV2: P13, L4-5: This is confusing: "The time contact of air with sea-ice shown by backward trajectory was twice of 10 hours, twice of 20 hours, and once of 60 hours"

AUT: Right the wording was corrected as "The contact time of air with sea-ice shown by backward trajectory was twice 10 hours, twice 20 hours, and once 60 hours.

REV2: Section 5: This section should discuss the differences in the area of the topographic features that channel air from the interior to the coastal areas. Each of the stations may have distinctly difference source regions associated with local topography.

AUT: We agree and we introduce a new figure to discuss this point (second part of Figure 1). We also added the following: "These differences suggest that ozone rich air masses present over the inland Antarctic plateau are more efficiently transported to DDU than to NM and HA. Fig. 1 shows that the near-surface airflow between the Antarctic plateau and the coastal regions is largely controlled by the topography of the underlying ice sheets and the vicinity of low pressure systems at the coast of the Antarctic continent (Parish and Bromwich, 2007). Consequently, the airflow coming from inland Antarctica is important at DDU and to a lesser extent at SY, but not at NM and HA. Âż

REV2: P12, L12,L20: "downslope" is more general. Low pressure systems on the coast can produce similar downslope conditions. It would be interesting to do a composite synoptic map for high and low ozone conditions at DDU etc.

AUT: Right we now report backward trajectories traces at DDU and discuss the role of topography and surface wind with the help of the second part of Figure 1.

REV2: Section 6. This section is rather speculative and could be tightened up considerably.

AUT: We discarded the discussion of Concordia data with respect to stratospheric change (SAOZ data initially reported in Table 1). Nevertheless since we found an opposite trend to the previous one published by Helmig, the difference has to be discussed a bit. At the end, this section was shortened as far as possible.

---

## Author Response (AR1)

General: This is a reasonably constructed paper largely focused on comparison of ozone measurements over the high plateau versus those in coastal areas over a period of less than a decade. A significant amount of text addressed ozone production versus loss terms in the context of several case studies from balloon profiling. Section 5, describing observations at other sites on the coasts, could be in Supplemental material so that the paper stays focused on high plateau (DC and SP) and provides a comparison with a coastal site (DDU). Well, as stated in the introduction, the paper aims to fill the lack of ozone data in East Antarctica (both central and coastal). Also, the OPALE project addressed the topics of oxidants at the central and coastal sites in East Antarctica. The comparison of the different coastal sites is also of great interest and thus we think that this coastal aspect is not at all marginal in the paper.

In a number of cases, it was hard to identify the "bottom line" or how robust the conclusions were. In terms of trends, in the face of decadal variability in the circulation around Antarctica, it would be useful to show some composite synoptic maps (e.g. high and low ozone at DDU) and assess if the patterns have changed. In some of the discussion, it was easy to get lost in the details. It would also be nice to have some plots from ERA-I showing synoptic patterns rather than abstract statements of trajectories originating from x, y, or z (I looked at vector wind plots and maps of GPH for some of the examples.). OK in the revised version we report backward trajectories in summer (high and low ozone values) both for DDU and Concordia to better highlight the role of the transport on ozone levels (in addition to fig. 5 and 10).

Specific areas for improvement: P4, L9 and L12: "Adjustment" or "change" are probably better words than "correction." We think that correction is correct in this context.

P6, L7-20: The trajectory length analysis should be supported by a cluster plot showing the actual trajectories (or trajectory origins) overlain on the Antarctic region. This might help identify preferred source areas, if any. For example, do the trajectories follow terrain isopleths to DC or do they originate over the high terrain around Dome A? This could be a supplemental figure. As above mentioned we introduced a figure showing traces of backward trajectories from which region air masses reaching Concordia are coming from.

P6, L21-23: This deserves some discussion of the high NOx fluxes in early December reported by Frey et al and the high surface nitrate at that time vis-à-vis the NOx-to-O3 conversion process. It is also unfortunate that the Gallee et al. modeling work for OPALE did not include O3 calculations. We agree

P6, L26+: Table I displays total column ozone with no discussion of its importance. Also there needs to be a few words explaining the red highlighted O3 values (the highest values in each time period. This Table (including the SAOZ values now not discussed in section 6, see below) was removed. Instead we report backward trajectories for the month of December (2007 to 2014) (a new figure, Fig. 6 has been introduced) to generalize the role of the transport on ozone values as shown for the Figure 4 and 5.

P6: Was there any evidence of stratospheric intrusions. Although rare, Crawford et al 2001 showed one case in early December with an increase in 7Be while Traversi et al 2014 (Tellus Series B-Chemical and Physical Meteorology) argued for such a signature in nitrate

measurements at Concordia. At least for summer, vertical ozone soundings from either the South Pole or Concordia do not reveal an importance of stratospheric intrusion for the surface ozone budget. We think that arguments developed by Traversi on nitrate are not directly helpful for the ozone concern (ozone is gaseous species, nitrate coming from the stratospheric reservoir would partly be due to PSCs sedimentation, Be is stuck on sub-micronic sulphate, ….).

P7, L10: Fig. 5a only shows length of trajectory not its origin. The claim of origin on the "high plateau" needs to be supported, especially where on the high plateau. Right and in the new version we removed Table 1 that was not very useful and instead we plotted all backward trajectories for December (over the 7-8 years of the record) as a function of the corresponding ozone range. This figure far better illustrates the importance of the transport.

Section 3.1.2: The discussion of cases examining the competing effects of ozone production versus dilution in a growing convective boundary took a bit to follow before a final conclusion that dilution dominated the afternoon drop of ozone. The amount of data does not justify any robust conclusions. This should be emphasized. We agree and reworded the text as follows: Though further investigations with diurnal vertical profiling over the course of the day are clearly needed, this example suggests that the decrease of ozone mixing ratio often detected near the surface in the afternoon (Fig. 8) may be related to dilution from the increase in the boundary layer depth which counteracts the local ozone photochemical production."

The authors should also look at the changing meteorology 1-4 January 2010 as a strong ridge developed over DC by 3 January (I looked at ERA-I results). The associated subsidence might account for the shallow mixing layer on 3 January compared to 31 December. We agree but it should be at the border of the scope of this paper to describe synoptic conditions associated with day to day change of the vertical profiles.

P11, L9: ". . .difference in the dynamic of the lower atmosphere. . ." is vague. Right, we added in parenthesis (more stratified lower atmospheric layers at the South Pole than at Concordia).

P13, L4-5: This is confusing: "The time contact of air with sea-ice shown by backward trajectory was twice of 10 hours, twice of 20 hours, and once of 60 hours" Right the wording was corrected as "The contact time of air with sea-ice shown by backward trajectory was twice 10 hours, twice 20 hours, and once 60 hours.

Section 5: This section should discuss the differences in the area of the topographic features that channel air from the interior to the coastal areas. Each of the stations may have distinctly difference source regions associated with local topography. We agree and we introduce a new figure to discuss this point (second part of Figure 1). We also added the following: "These differences suggest that ozone rich air masses present over the inland Antarctic plateau are more efficiently transported to DDU than to NM and HA. Fig. 1 shows that the near-surface airflow between the Antarctic plateau and the coastal regions is largely controlled by the topography of the underlying ice sheets and the vicinity of low pressure systems at the coast of the Antarctic continent (Parish and Bromwich, 2007). Consequently, the airflow coming from inland Antarctica is important at DDU and to a lesser extent at SY, but not at NM and HA. »

P12, L12,L20: "downslope" is more general. Low pressure systems on the coast can produce similar downslope conditions. It would be interesting to do a composite synoptic map for high and low ozone conditions at DDU etc. Right we now report backward trajectories traces at DDU and discuss the role of topography and surface wind with the help of the second part of Figure 1.

Section 6. This section is rather speculative and could be tightened up considerably. We discarded the discussion of Concordia data with respect to stratospheric change (SAOZ data initially reported in Table 1). Nevertheless since we found an opposite trend to the previous one published by Helmig, the difference has to be discussed a bit. At the end, this section was shortened as far as possible.

**Anonymous Referee #3**

General Comments: This manuscript provides a nice synopsis of multi-year surface ozone records at two Antarctic locations: almost 8 years at the inland Concordia Station, and almost 11 years at the coastal Dumont d'Urville Station. A comparative analysis is presented between these stations, South Pole, Syowa, Neumayer and Halley, and the role of various factors influencing the characteristics of the annual and diurnal cycles of surface ozone, such as topography, meteorology, proximity to the ocean, and oxidant chemistry, is examined. With several relatively minor adjustments, this manuscript will be appropriate for publication in ACP.

The most significant issue I would like to see addressed is analysis presented in section 3.1.1. on the relationship between trajectories and ozone concentrations. This is rather unsatisfying, since it is based on a "pick and choose" approach instead of the analysis of patterns that can be rigorously generalized. It also places more trust in trajectory models than they probably deserve, particularly beyond 4-5 days out. I took the liberty of plotting the data in the first two columns of Table 1 (see attached figure). There is no need for me to expand on this plot, since it says everything that needs to be said. It is of course true that there appear to be some correlations in Fig. 4 if some periods are inspected in isolation, but this method is scientifically not terribly convincing. Thank you for this comment: indeed Table 1 was not very convincing since everything was smoothed when averaged over one month or half a month. So we remove it and instead we simply plotted all backward trajectories for December (over the 7-8 years of the record) as a function of the corresponding ozone range. This figure far better illustrates the importance of the transport.

An analysis along the lines of what was presented in Bottenheim and Chan (2006) may, at least partially, alleviate these concerns. With 8 and 11 years of data, there should be enough material here that this type of analysis would provide further insights. That is indeed another possibility but we think that this would be less straightforward in Antarctica than in the case of Arctic. Anyway with the figure of backward trajectories introduced in the new version, the role of transport is clearly seen.

Specific Comments:

Page 1 Line 24: cycles: Done
P1L26: but not at the South Pole, and a … This sentence needs to be rephrased. Do you mean "far better mixed layer with higher ozone concentrations"? Also, the 250m needs to be reconciled with numbers in section 4.

This long sentence was reworded into two sentences as follows: "However, in summer the diurnal cycle of ozone is different at the two sites with a drop of ozone in the afternoon at Concordia but not at the South Pole. The vertical distribution of ozone above the snow surface also differs. When present, the ozone rich layer located near the ground is better mixed and deeper at Concordia (up to 400 m) than at the South Pole during sunlight hours."

P2L5: research; separate the effect Done
P2L15: remove "that" Done
P2L22: (Crawford... Done
P3L18: papers in Done
P3L32: at DDU Done

P4L1: were made once a month Done

P4L21: treat McClure et al like any other reference: just give the authors and date, the rest goes to the end of the paper Done

P4L28: prior to Done

P4L31: the upward shift: if the delay time is predictable, then the shift can be corrected. Was this done to finalize the data? As mentioned in the text the shift is variable from one flight to another flight but never exceeded 20 m. The effect remains weak for the cases discussed in section 3.1.2.

P4L33: This permitted the documentation of the ... P5L2: proved Done

Section 2 should also briefly explain SAOZ (which is used in Table 1 but never spelled out until the Acknowledgements). OK, since Table 1 was removed following the major issue rise at the beginning of your review, and following the recommendation of the other reviewer to reduce the discussion in Section 6 (restricting the discussion on summer trends to the South Pole) the SAOZ data at Concordia are not anymore needed.

And it needs to include some details on the trajectory model used. These models are not absolute, and some are better than others. 5 days out, most of them have rather large uncertainties.

Right and we add a paragraph at the end of section 2 as follows: "To characterize the origin of air masses reaching the DDU and Concordia regions, five day backward trajectories were computed using the Hybrid Single-Particle Lagrangian Integrated Trajectory model (R. R. Draxler and G. D. Rolph, NOAA Air Resources Laboratory, Silver Spring, Maryland, 2003, available at http://www. arl.noaa.gov/ready/hysplit4.html). Meteorological data from Global Data Assimilation Process (available at ftp://arlftp. arlhq.noaa.gov/pub/archives/gdas1) were used as input, and the model was run every 6 h in backward mode for three different altitudes (0, 250, and 500 m agl)."

I think p.6 can be shortened to one paragraph, and table 1 should be eliminated, because it does not provide convincing evidence of any patterns, even if the patterns the authors strain to see make physical sense.

Thank you for this comment. We agree and discarded this Table (including the SAOZ values now not discussed in section 6). Instead, we report traces of backward trajectories for the month of December (2007 to 2014) (a new figure, Fig. 6 has been introduced) to generalize the role of the transport on ozone values as also shown in the Figure 4 and 5 (but for specific time periods).

The text was updated as follows: "Even on the monthly scale, during the summer the influence of the air mass trajectory on ozone levels is still visible. For instance, as seen in Fig. 6 for December, the highest ozone values (higher than 38 ppbv) correspond to the shortest trajectories that almost always spent a significant time fraction over the highest part of the plateau (above 3500 m asl). Trajectories corresponding to ozone values ranging between 32 and 38 ppbv travelled mostly above 2500 m asl in the inner part of the continent. Finally, ozone values below 26 ppbv are observed when the trajectories were among the longest, sometimes starting from coastal regions or from the ocean (Fig. 6)."

P7L10: invariant instead of "not modified". Done

P7L26: remove "neutral"; convective conditions would conceivably also produce a straight

profile Right, Done
P8L1: no longer Done
P8L2: generally several hundreds of m deep as seen... Done
P8L7: radiation Done
P8L32: Details Done
P9L6: remove "very" Done
P9L17: dependent Done
P8L23: lasted to the Done
P9L30: remove "over" Done
P10L19: absence of detectable increase: what about the potential role of meteorology / mixing? Yes that is possible but we cannot address this point (as we did for Concordia) since deploying tethered balloons is far more complicated at DDU compared to Concordia (more wind and turbulent regime).

Also, I suspect the diurnal cycle of O3 at Concordia isn't always as nice as that shown in Fig. 7. You are right and this was explained at the beginning of section 3.1.2: only under very specific conditions is a diurnal cycle: "Given the importance of synoptic transport conditions on the day-to-day ozone variability (Sect. 3.1.1), the detection of an ozone trend over several days to estimate a mean local ozone production rate requires the search for a period of several days over which the transport pattern was invariant. As seen in Fig. 5a, from 27 to 30 November 2009 backward trajectories indicate transport of air masses consistently from the high Antarctic plateau (100% of time above 3200 m asl), and over these 4 days surface wind direction remained unchanged (Fig. 8).

P11L5: radiation... acts Done

P11L9: dynamics Done

P11L27: the "always" seems to contradict the statement in L19 about 25% of profiles not having significant vertical gradients; No since in fact here we are discussing the cases (75%) when a vertical gradient is present.
P13L5: remove the "of"s Done

P14L20: On the other hand, Done

P15L7: of diurnal, Done
P15L8: which are, Done
P15l15: On the other hand, Done

The Conclusions should also mention the winter trend. This is interesting as well. Right and we added "Finally, the DDU data indicate a significant increasing trend in winter, motivating the extension of time series in view to examine the possible influence of stratosphere-troposphere transport for instance."

Table 1: should be removed (see comments above). If you really wish to keep it, you need to discuss the SAOZ column somewhere in the text (which is currently not the case), and explain the colour code (red/blue/black numbers). As explained above we follow your suggestion to remove this Table. We also discard from the discussion the SAOZ data to shorten the discussions in Sect. 6.

Table 2: use standard exponential notation: OK Done

Table 3: give the heights in "agl" please, so they can be related to what's in the text: OK Done

Fig. 4: identify the interesting episodes discussed in the text with arrows: Done
Fig. 10: a grey arrow is mentioned in the caption, but I can' find it. Thank you for this remark, in fact the caption was wrong. We now add arrows to highlight episodes discussed in the text.

[revised manuscript text omitted]

Michel Legrand 18/5/y 11:23

Michel Legrand 14/5/y 16:22

Michel Legrand 18/5/y 11:23